# CoDi: Subject-Consistent and Pose-Diverse Text-to-Image Generation

**Zhanxin Gao**[1]  **Beier Zhu**[2]  **Liang Yao**[3]  **Jian Yang**[1]  **Ying Tai**[1]*

[1]Nanjing University, [2]University of Science and Technology of China, [3]Vipshop

zxgao@smail.nju.edu.cn  beier.zhu@ustc.edu.cn  yingtai@nju.edu.cn

## Abstract

Subject-consistent generation (SCG)—aiming to maintain a consistent subject identity across diverse scenes—remains a challenge for text-to-image (T2I) models. Existing training-free SCG methods often achieve consistency at the cost of layout and pose diversity, hindering expressive visual storytelling. To address the limitation, we propose subject-Consistent and pose-Diverse T2I framework, dubbed as CoDi, that enables consistent subject generation with diverse pose and layout. Motivated by the progressive nature of diffusion, where coarse structures emerge early and fine details are refined later, CoDi adopts a two-stage strategy: Identity Transport (IT) and Identity Refinement (IR). IT operates in the early denoising steps, using optimal transport to transfer identity features to each target image in a pose-aware manner. This promotes subject consistency while preserving pose diversity. IR is applied in the later denoising steps, selecting the most salient identity features to further refine subject details. Extensive qualitative and quantitative results on subject consistency, pose diversity, and prompt fidelity demonstrate that CoDi achieves both better visual perception and stronger performance across all metrics. The code is provided in https://github.com/NJU-PCALab/CoDi.

## 1 Introduction

While text-to-image (T2I) (Ramesh et al., 2022; Saharia et al., 2022; Rombach et al., 2022; Blattmann et al., 2023) models excel in high-quality image generation (Rombach et al., 2022; Mou et al., 2024), they struggle to maintain subject consistency across multiple scenes. Subject-consistent generation (SCG) aims to synthesize images of the same subject across diverse contextual prompts with three key objectives: (1) ensuring subject consistency across generated instances, (2) promoting layout and pose diversity across different instances to avoid repetitive or overly similar compositions, and (3) maintaining prompt fidelity to accurately reflect the semantics of each prompt. The capability enables numerous practical applications including multi-scene narrative for visual storytelling, customizable character design for animation and gaming, and coherent illustration sequences for graphic novels.

Current SCG methods (Kopiczko et al., 2024; Ye et al., 2023) primarily rely on training-intensive optimization (Avrahami et al., 2024) or mapping networks (Ruiz et al., 2024; Gal et al., 2023b) to bind subjects to latent representations. These approaches often require computationally expensive fine-tuning per subject or depend on domain-specific encoders, limiting scalability and generalizability. Training-free methods (Tewel et al., 2024; Zhou et al., 2024) have gained significant attention due to their elimination of parameter tuning, strong generalization capabilities, and broad compatibility with diverse diffusion architectures. Current training-free methods—ConsiStory (Tewel et al., 2024) and StoryDiffusion (Zhou et al., 2024)—enhance subject consistency by sharing self-attention keys and values across generated images. However, as noted in their limitations (Tewel et al., 2024; Hertz et al., 2024) and evident in Fig. 1, these methods often achieve high consistency at the cost of severely reduced layout and pose diversity, making it challenging to balance all three objectives.

To better balance the three objectives, we propose a training-free framework—subject-Consistent and pose-Diverse generation, dubbed CoDi—that achieves strong subject consistency while preserving diverse poses. Motivated by the progressive nature of diffusion models (Yue et al., 2024)—which

---

*indicates corresponding author.

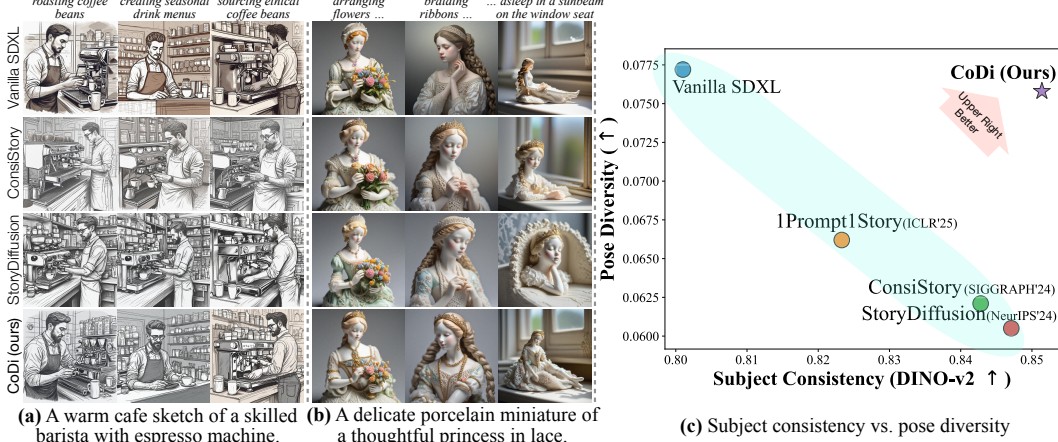

(a) A warm cafe sketch of a skilled barista with espresso machine.

(b) A delicate porcelain miniature of a thoughtful princess in lace.

(c) Subject consistency vs. pose diversity

Figure 1: Comparison of subject-consistent generation methods: Vanilla SDXL (Podell et al., 2023), ConsiStory (Tewel et al., 2024), StoryDiffusion (Zhou et al., 2024) and CoDi (ours). **(a&b)** Existing methods sacrifice pose diversity for subject consistency, *e.g.*, ConsiStory produces similar poses in Figure 1 (a); and the lower right with hands placed in front in Figure 1 (b). In contrast, CoDi generates consistent subjects, while matching the pose diversity of Vanilla SDXL. **(c)** Subject consistency vs. pose diversity. Current methods struggle to balance the two, whereas CoDi achieves both effectively.

shows that low-frequency attributes like pose and layout are formed in early denoising steps, while high-frequency details such as facial features emerge later—our CoDi adopts a two-stage strategy: Identity Transport (IT) and Identity Refinement (IR). During the early denoising steps, IT uses optimal transport to align each target image's features with the reference identity features. Intuitively, this resembles mosaicking: assembling the subject using visual pieces from the reference image, rearranged to match the target pose—thus naturally preserving identity and keeping the original pose. In the later denoising steps, IR further refines subject consistency by guiding each target image to attend to the most salient identity attributes via cross-attention. As shown in Figure 1, CoDi achieves superior visual results in both subject consistency and pose diversity, and quantitatively demonstrates advantages over existing methods in balancing this trade-off.

We evaluate our method on the existing T2I SCG benchmark ConsiStory+ (Liu et al., 2025). Compared to other training-free approaches, both quantitative and qualitative results validate that our framework achieves better subject consistency while preserving richer layout and pose diversity. It demonstrates a superior trade-off among subject consistency, pose diversity, and prompt fidelity. Further analysis is also provided to demonstrate CoDi's advantages in pose diversity.

## 2 RELATED WORK

To steer T2I generation with diffusion models (Rombach et al., 2022; Podell et al., 2023; Nan et al., 2025; Esser et al., 2024; Zhu et al., 2025a; Wang et al., 2025a; Hu et al., 2023; Lin et al., 2025a; Zhou et al., 2025; Lin et al., 2025b; Wang et al., 2025b), various methods have been proposed to incorporate control signals such as depth maps, edge maps, and segmentation (Mei et al., 2025; Zhang et al., 2023; Yang et al., 2023; Lei et al., 2025; Chen et al., 2025). Among them, subject consistency (a.k.a identity preservation) has attracted growing attention, aiming to generate a set of images conditioned on a specified subject. Existing subject-consistent generation (SCG) methods can be broadly categorized into two groups: training-based and training-free.

**Training-based SCG.** Training-based methods require either (1) fine-tuning on additional training data (Yang et al., 2024; Li et al., 2024; 2019; Betker et al., 2023; Liu et al., 2024) or (2) test-time optimization using reference images (Roich et al., 2022; Gal et al., 2023b; Kumari et al., 2023; Xiao et al., 2024). The first line of work, represented by StoryDALL-E (Maharana et al., 2022) and Make-A-Story (Rahman et al., 2023), incorporates additional modules to capture subject information, followed by fine-tuning on large datasets to enable direct control over the subject given a reference image. The second line of work, exemplified by DreamBooth (Ruiz et al., 2023) and Textual Inversion (Gal

et al., 2023a), optimizes model parameters or token embeddings on the given test images to inject subject identity. Despite their success in maintaining subject consistency, training-based methods suffer from high training costs or significant test-time latency. In contrast, our `CoDi` is training-free and introduces only mild additional latency.

**Training-free SCG.** Training-free methods circumvent the need for iterative tuning of model parameters. For instance, 1Prompt1Story (Liu et al., 2025) improves consistency by aligning prompt embeddings across generations. However, textual embedding control alone is insufficient to to enforce consistency, often resulting in subject drift. The current leading methods, ConsiStory (Tewel et al., 2024) and StoryDiffusion (Zhou et al., 2024), adopt attention-based mechanisms to promote subject consistency by sharing self-attention keys and values. However, as noted in their limitation discussions (Tewel et al., 2024; Hertz et al., 2024), *applying attention across a set of images reduces pose diversity*. To address this issue, our `CoDi` explicitly preserves diversity and promotes consistency by aligning early-stage features between the target and reference images via optimal transport.

## 3 METHOD

Our `CoDi` consists of two stages: Identity Transport (`IT`) and Identity Refinement (`IR`). Our `IT` operates in the early denoising stage to transport identity features from the reference image while preserving the pose and background of the target images. `IR` is applied in later denoising stages to refine subject consistency in fine-grained details. This two-stage design is inspired by (Yue et al., 2024), which reveals that low-frequency attributes such as pose and layout are determined early in the denoising timesteps, whereas high-frequency components like facial details emerge in later steps. We begin with the setup of subject-consistent generation (SCG), a review of attention-based SCG methods and a brief introduction of optimal transport.

### 3.1 PRELIMINARIES

**Setup.** SCG aims to synthesize a batch of images that share the same subject identity across diverse scenes. Formally, given a set of $N$ textual prompts $\{\mathbf{t}_n\}_{n=1}^N$, where each prompt is composed of a shared identity prompt $\mathbf{t}_{\mathsf{id}}$ and a unique attribute prompt $\mathbf{a}_n$, *i.e.*, $\mathbf{t}_n = [\mathbf{t}_{\mathsf{id}}, \mathbf{a}_n]$. For instance, given $\mathbf{t}_1$ = "A hyper-realistic digital painting of a fairy giggling in a grove of enchanted crystals" and $\mathbf{t}_2$ = "A hyper-realistic digital painting of a fairy lost in a maze of giant sunflowers", the identity prompt is $\mathbf{t}_{\mathsf{id}}$ = "a hyper-realistic digital painting of a fairy", and the attribute prompts are $\mathbf{a}_1$ = "giggling in a grove of enchanted crystals" and $\mathbf{a}_2$ = "lost in a maze of giant sunflowers". We refer to the image generated from the identity prompt $\mathbf{t}_{\mathsf{id}}$ as the *reference image*, denoted as $\mathbf{x}_{\mathsf{id}}$. The objective is to generate *target images* $\{\mathbf{x}_n\}_{n=1}^N$ that depict a visually consistent subject with $\mathbf{x}_{\mathsf{id}}$, while capturing the scene-specific attributes described in $\mathbf{a}_n$. See Figure 2 for a concrete example.

**Review of cross-image attention SCG.** The current leading training-free SCG methods, ConsiStory (Tewel et al., 2024) and StoryDiffusion (Zhou et al., 2024), adopt attention-based strategies that extend the standard self-attention to cross-image attention mechanism. Formally, let $\{X_n\}_{n=1}^N$ denote the features of the target images $\{\mathbf{x}_n\}_{n=1}^N$. For generating $i$-th image, standard self-attention first projects $X_i$ to queries $Q_i$, keys $K_i$, and values $V_i$, then compute

$$Z_i = \mathrm{Attn}(Q_i, K_i, V_i) = \mathrm{softmax}\left(\frac{Q_i K_i^\top}{\sqrt{d}}\right) V_i, \tag{1}$$

where $d$ is the feature dimension. Let $\oplus$ denote matrix concatenation. We compute the concatenated keys and values as $K_{1:N} = [K_1 \oplus ... \oplus K_N]$ and $V_{1:N} = [V_1 \oplus ... \oplus V_N]$, respectively. To enhance consistency, cross-image attention mechanism allows the feature of the $i$-th image, $X_i$, to attend to the values $V_{1:N}$ of other images using their corresponding keys $K_{1:N}$.

$$Z_i = \mathrm{Attn}(Q_i, K_{1:N}, V_{1:N}) = \mathrm{softmax}\left(\frac{Q_i K_{1:N}^\top}{\sqrt{d}}\right) V_{1:N}. \tag{2}$$

While both SCG methods adopt cross-image attention, they differ slightly in implementation: ConsiStory (Tewel et al., 2024) limits attention to masked subject regions, whereas StoryDiffusion (Zhou et al., 2024) randomly samples tokens from all regions without subject constraints.

$\mathbf{t}_{\mathsf{id}}$: a hyper-realistic digital painting of a fairy.    $\mathbf{t}_n$: a hyper-realistic digital painting of a fairy giggling in a grove of enchanted crystals.

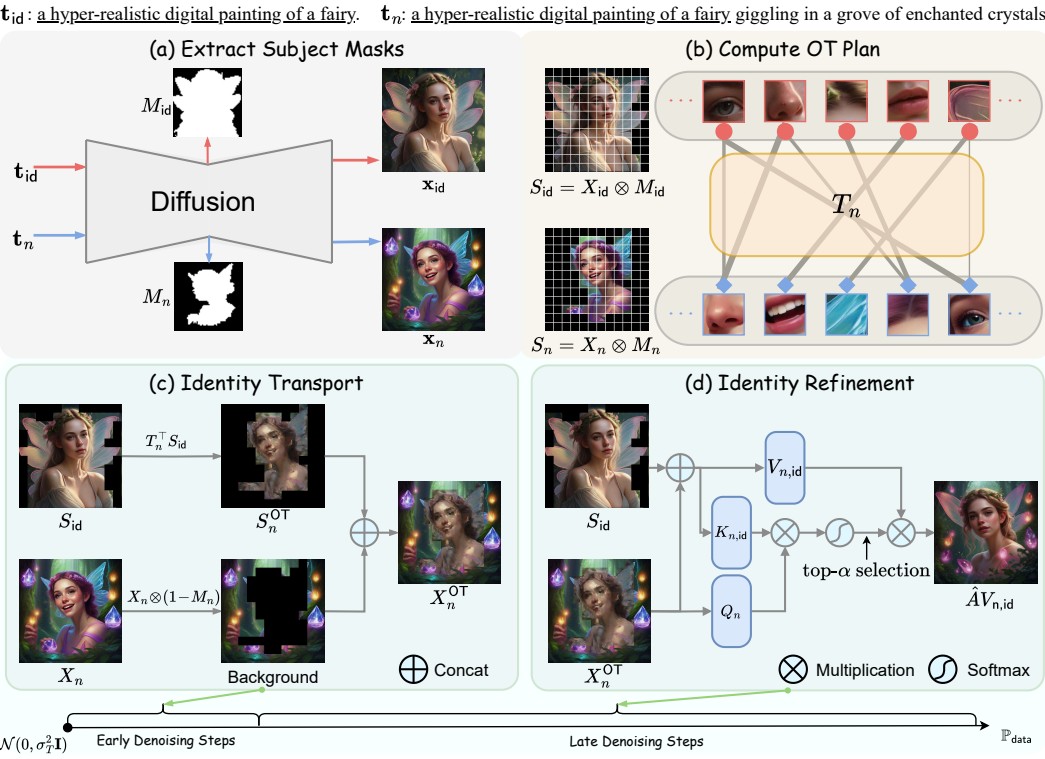

Figure 2: Illustration of our CoDi. **(a)** Extract subject masks ($M_{\mathsf{id}}$ and $M_n$) by averaging the image-text cross-attention at the final denoising timestep for subject-related tokens (*e.g.*, "fairy"). **(b)** Compute the OT plan $T_n$ using the cost matrix $C$ and the probability masses **a** and **b** (detailed in Sec. 3.2). **(c)** Identity transport (IT) operates in the early denoising steps to transfer reference subject features to targe images in a pose-aware manner. **(d)** Identity refinement (IR) operates in the late denoising steps to refine subject details using selective cross-image attention mechanism.

As discussed in their limitations (Tewel et al., 2024; Hertz et al., 2024), attention-based methods significantly reduce layout and pose diversity. We conjecture that attending to a shared pool of keys and values *entangles feature updates across images, implicitly aligning spatial layouts and poses.* To mitigate this, prior work (Tewel et al., 2024) introduces components such as attention dropout and query blending. However, these additions increase computational overhead and still fail to recover pose diversity (as shown in Figure 1). In this paper, we draw inspiration from structural learning to simultaneously preserve subject consistency and pose diversity by transporting identity features to each target image via optimal transport.

**Optimal transport.** Optimal transport (OT) (Villani et al., 2008; Monge, 1781; Zhu et al., 2025b) provides a framework for measuring the distance between two distributions. Specifically, given two sets of support features $\{\mathbf{v}_m\}_{m=1}^{M}$ and $\{\mathbf{u}_n\}_{n=1}^{N}$, we define two discrete distributions $\mathbb{P}$ and $\mathbb{Q}$ as:[1]

$$\mathbb{P}(\mathbf{x}) = \sum_{m=1}^{M} a_m \delta(\mathbf{v}_m - \mathbf{x}), \quad \mathbb{Q}(\mathbf{x}) = \sum_{n=1}^{N} b_n \delta(\mathbf{u}_n - \mathbf{x}) \tag{3}$$

where $\delta(\cdot)$ denotes the Dirac function, and $a_m$, $b_n$ denote the associated probabilities that sum to 1, respectively. Let $\mathbf{a} = [a_1, ..., a_M]^\top$ and $\mathbf{b} = [b_1, ..., b_N]^\top$. Given a cost matrix $C \in \mathbb{R}^{M \times N}$, where each entry $C(m, n)$ denotes the transport cost between $\mathbf{v}_m$ and $\mathbf{u}_n$ (typically defined by their similarity), the OT distance between $\mathbb{P}$ and $\mathbb{Q}$ is defined as:

$$d_{\mathsf{OT}}(\mathbb{P}, \mathbb{Q}; C) = \min_{T \geq 0} \langle T, C \rangle, \text{ s.t. } T\mathbf{1}_M = \mathbf{a}, T^\top \mathbf{1}_N = \mathbf{b}, \tag{4}$$

where $T \in \mathbb{R}^{M \times N}$ is the transport plan, with $T(m, n) \geq 0$ representing the amount of mass moved from $\mathbf{v}_m$ to $\mathbf{u}_n$, $\langle, \rangle$ denotes the Frobenius inner product, $\mathbf{1}_M$ is $M$-dimensional all-one vector.

---

[1] We slightly abuse the notations $\mathbf{x}$ and $N$, which here do not refer to an image or the number of target images.

## 3.2 IDENTITY TRANSPORT

Our IT operates in the early denoising steps (*e.g.*, the first 10 of 50 total steps) to independently transport identity features from the reference image $\mathbf{x}_{\mathsf{id}}$ to each target image $\mathbf{x}_n$ for all $n \in [1, N]$. Our IT begins by extracting subject features from masked regions.

**Extract subject features.** Masking out background regions offers two benefits for subject consistency: it reduces background interference and computational cost by focusing on the subject alone. We adopt a similar strategy to that of previous methods (Hertz et al., 2023; Tewel et al., 2024), using image-text cross-attention to extract subject masks. Specifically, let $X_{\mathsf{id}}$ denote the features of the reference image $\mathbf{x}_{\mathsf{id}}$ generated from the identity prompt $\mathbf{t}_{\mathsf{id}}$. When generating $\mathbf{x}_{\mathsf{id}}$, we average the cross-attention maps at the final denoising timestep for subject-related tokens (*e.g.*, "fairy"), followed by applying Otsu's method (Otsu et al., 1975) to produce a binary mask $M_{\mathsf{id}}$. This mask highlights the subject-relevant regions, from which we extract the subject features as:

$$S_{\mathsf{id}} = X_{\mathsf{id}} \otimes M_{\mathsf{id}} \in \mathbb{R}^{s_{\mathsf{id}} \times d} \tag{5}$$

where $\otimes$ applies the binary mask to retain subject features, $s_{\mathsf{id}}$ denotes the number of ones in the binary mask $M_{\mathsf{id}}$, and $d$ is the feature dimension. Similarly, for each target image $\mathbf{x}_n$, we extract subject features as $S_n = X_n \otimes M_n \in \mathbb{R}^{s_n \times d}$. The process is visualized in Figure 2 (a).

**Transport between $S_{\mathsf{id}}$ and $S_n$.** Given the subject feature pairs $S_{\mathsf{id}} = [\mathbf{s}_{\mathsf{id}}^1, ..., \mathbf{s}_{\mathsf{id}}^{s_{\mathsf{id}}}]^\top$ and $S_n = [\mathbf{s}_n^1, ..., \mathbf{s}_n^{s_n}]^\top$, we first derive an optimal transport plan $T$ that aligns the reference features set $\{\mathbf{s}_{\mathsf{id}}^i\}_{i=1}^{s_{\mathsf{id}}}$ with the target features $\{\mathbf{s}_n^i\}_{i=1}^{s_n}$ (See Figure 2 (b)). Using this plan $T$, we compose the target subject features by transporting features from the reference image. Intuitively, this process resembles **mosaicking**: we assemble the target subject using pieces from the reference image, **rearranged to match the target pose**. Since the visual pieces originate from the reference image, **subject identity is naturally preserved**. To solve the OT problem in Eq. (4), we first define the cost matrix $C$ and the associated probability masses $\mathbf{a}$ and $\mathbf{b}$.

*Definition of the cost matrix $C$.* The cost matrix is typically defined based on the pairwise distances between features: smaller distances imply lower transport costs. For a pair $\mathbf{s}_{\mathsf{id}}^i$ and $\mathbf{s}_n^j$ from final denoising step (where features contain minimal noise), the cost is defined as:

$$C(i, j) = 1 - \cos(\mathbf{s}_{\mathsf{id}}^i, \mathbf{s}_n^j) = 1 - \frac{\mathbf{s}_{\mathsf{id}}^{i\top} \mathbf{s}_n^j}{\|\mathbf{s}_{\mathsf{id}}^i\|_2 \|\mathbf{s}_n^j\|_2}. \tag{6}$$

*Definition of the probability masses $\mathbf{a}$ and $\mathbf{b}$.* Intuitively, $\mathbf{a} = [a_1, \ldots, a_{s_{\mathsf{id}}}]^\top$ represents the importance weights of the subject features, where a larger $a_i$ indicates that feature $\mathbf{s}_{\mathsf{id}}^i$ is more relevant to the subject $\mathbf{t}_{\mathsf{id}}$. We reuse the average cross-attention maps for generating the subject-relevant mask as the feature importance and apply softmax function to ensure the sum $\sum_i a_i$ equals to 1. The importance weights $\mathbf{b}$ for the target subject features $S_n$ are derived analogously.

With the cost matrix $C$ and the probability masses $\mathbf{a}$ and $\mathbf{b}$, we solve the OT plan $T_n$ in Eq. (4) using network simplex algorithm (Orlin, 1997). With the derived $T_n$, the subject target features composed by reference subject features are computed as

$$S_n^{\mathsf{OT}} = T_n^\top S_{\mathsf{id}}. \tag{7}$$

To form the final representation $X_n^{\mathsf{OT}}$, we combine $S_n^{\mathsf{OT}}$ with the non-subject features (masked out by $M_n$) from $X_n$. The representation is then passed through the diffusion network to produce the output. The IT process is illustrated in Figure 2 (c).

## 3.3 IDENTITY REFINEMENT

The motivation behind this stage is that the IT module performs a coarse transport between $S_{\mathsf{id}}$ and $S_n$. However, since the binary subject masks are imprecise and the foreground of target images evolves during denoising—while our transport plan $T_n$ remains fixed—further refinement of subject details becomes necessary.

Our IR operates in the later denoising steps (*e.g.*, the last 40 of 50 total steps) to reinforce subject details in the target images. IR resembles cross-image attention-based SCG methods, except that

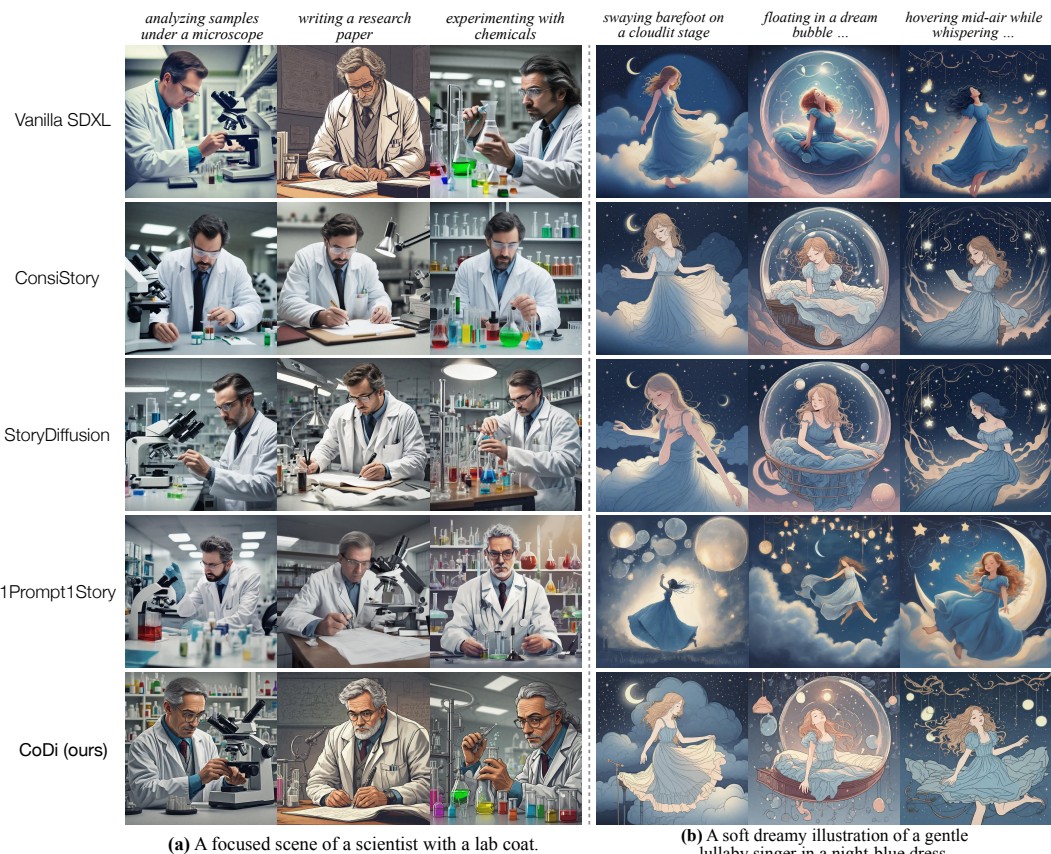

(a) A focused scene of a scientist with a lab coat.

(b) A soft dreamy illustration of a gentle lullaby singer in a night-blue dress.

Figure 3: **Qualitative comparison** among Vanilla SDXL (Podell et al., 2023), ConsiStory (Tewel et al., 2024), StoryDiffusion (Zhou et al., 2024), and 1Prompt1Story (Liu et al., 2025). ConsiStory and StoryDiffusion generate similar poses across examples, while 1Prompt1Story preserves pose diversity but struggles with subject consistency. In contrast, our CoDi achieves both.

each target image attend only to the most relevant reference features to avoid entangled feature update across target images. Specifically, to generate the $n$-th image, we first construct the concatenated keys and values as $K_{n,\text{id}} = [K_n \oplus K_{\text{id}}]$ and $V_{n,\text{id}} = [V_n \oplus V_{\text{id}}]$, respectively. The cross-image attention scores are compute as

$$A_n = \text{softmax}\left(\frac{Q_n K_{n,\text{id}}^\top}{\sqrt{d}}\right). \tag{8}$$

For each query, we retain only the top-$\alpha$ attention scores of the reference tokens (*i.e.*, $K_{\text{id}}$). Specifically, for each row $A_i$ of $A$, we define the top-$\alpha$ index set $\mathcal{I}_i$ (see Appendix A for details) and zero out all other entries of $K_{\text{id}}$:

$$\tilde{A}_{ij} = \begin{cases} A_{ij}, & \text{if } j \in \mathcal{I}_i \\ 0, & \text{otherwise} \end{cases} \quad \text{and} \quad \hat{A}_i = \frac{\tilde{A}_i}{\sum_{j \in \mathcal{I}_i} \tilde{A}_{ij}}. \tag{9}$$

The final cross-attention output is then computed as:

$$\text{Attn}_\alpha(Q_n, K_{n,\text{id}}, V_{n,\text{id}}) = \hat{A}V_{n,\text{id}} \tag{10}$$

This filtering mechanism ensures that only the most relevant identity features from the reference image contribute to the attention update. The IR process is demonstrated in Figure 2 (d).

Table 1: **Quantitative comparison** of subject consistency, pose diversity and prompt fidelity. Best results are marked in **bold**. ↑ indicates higher is better, and ↓ indicates lower is better.

| Method | Subject Consistency | | | Pose Diversity (↑) | Prompt Fidelity (↑) |
| | CLIP-I (↑) | DINO-v2 (↑) | DreamSim (↓) | | |
| --- | --- | --- | --- | --- | --- |
| Vanilla SDXL (Podell et al., 2023) | 0.8417 | 0.8010 | 0.3139 | 0.0772 | 0.9082 |
| 1Prompt1Story (Liu et al., 2025) | 0.8627 | 0.8233 | 0.2959 | 0.0662 | 0.8814 |
| ConsiStory (Tewel et al., 2024) | 0.8751 | 0.8428 | 0.2336 | 0.0621 | **0.9148** |
| StoryDiffusion (Zhou et al., 2024) | 0.8776 | 0.8471 | 0.2356 | 0.0605 | 0.9038 |
| CoDi (ours) | **0.8809** | **0.8514** | **0.2136** | **0.0758** | 0.9041 |

## 4 EXPERIMENTS

### 4.1 SETUP

**Benchmark.** We evaluate our CoDi on the standard SCG benchmark, ConsiStory+(Liu et al., 2025), which comprises nearly 200 prompt sets and supports the generation of over 1,100 images. Each prompt set includes a subject described in a specific style, with multiple frame-specific descriptions.

**Baselines and implementation details.** We compare our CoDi with SoTA training-free SCG methods, including ConsiStory (Tewel et al., 2024), StoryDiffusion (Zhou et al., 2024) and 1Prompt1Story (Liu et al., 2025). We reproduce all baselines using their official released code. All methods are implemented using the same backbone model, Stable Diffusion XL 1.0 (Podell et al., 2023), with an image resolution of $1024 \times 1024$, except for StoryDiffusion, which is evaluated at $768 \times 768$ due to its high memory consumption, following its original setting. To ensure fairness, identical noise seeds are enforced for all methods, ensuring that each prompt is initialized with the same random noise input. Hyperparameter $\alpha$ in Eq. (10) selects the top 50% of reference features.

**Evaluation metrics.** Our evaluation framework assesses the quality of generated images from three aspects: **(1)** subject consistency, **(2)** pose diversity, and **(3)** prompt fidelity. Subject consistency is evaluated by computing the average pairwise cosine similarity (or distance) between image embeddings within each target image set. We use three image encoders for this evaluation: CLIP-I (Hessel et al., 2021), DINO-v2 (Oquab et al., 2023), and DreamSim (Fu et al., 2023). To evaluate pose diversity, we extract 2D human joint coordinates using ViTPose's pose estimation model (Xu et al., 2022). To eliminate global variations in translation, rotation, and scale, we align poses using Procrustes analysis (Schönemann, 1966), inspired by standard practices in face alignment (Lin et al., 2021). The pose diversity score is then computed as the average Euclidean distance between corresponding keypoints across aligned image pairs. A higher score indicates greater pose diversity. For prompt fidelity, we use CLIP-Score (Hessel et al., 2021) to measure the cosine similarity between image and textual prompt embeddings. See Appendix B for more details.

### 4.2 EXPERIMENTAL RESULTS

**Qualitative comparison.** As shown in Figure 3, our CoDi achieves superior visual quality in terms of pose diversity, subject consistency, and prompt fidelity. Our CoDi preserves the pose diversity of Vanilla SDXL (Podell et al., 2023) while overcoming its limitation in subject consistency. In comparison, ConsiStory (Tewel et al., 2024) and StoryDiffusion (Zhou et al., 2024) achieve subject consistency at the cost of pose diversity. For example, in the scientist scenario, the man exhibits nearly identical body poses. Although 1Prompt1Story (Liu et al., 2025) maintains strong layout and pose diversity in both cases, its subject consistency remains limited.

**Quantitative comparison.** Table 1 presents a quantitative comparison. **(1)** Across all three subject consistency metrics—CLIP-I (Hessel et al., 2021), DINO-v2 (Oquab et al., 2023), and DreamSim (Fu et al., 2023)—our CoDi achieves the best performance, demonstrating superior identity preservation across instances. In particular, our method obtains the lowest DreamSim score (0.2136), indicating closer alignment with human perceptual similarity than competing methods. **(2)** In terms of pose diversity, CoDi achieves the highest score (0.0758), closely matching Vanilla SDXL (0.0772). This demonstrates its ability to preserve the inherent pose diversity of the diffusion model while maintaining subject consistency. **(3)** For prompt fidelity, CoDi performs competitively—ranking

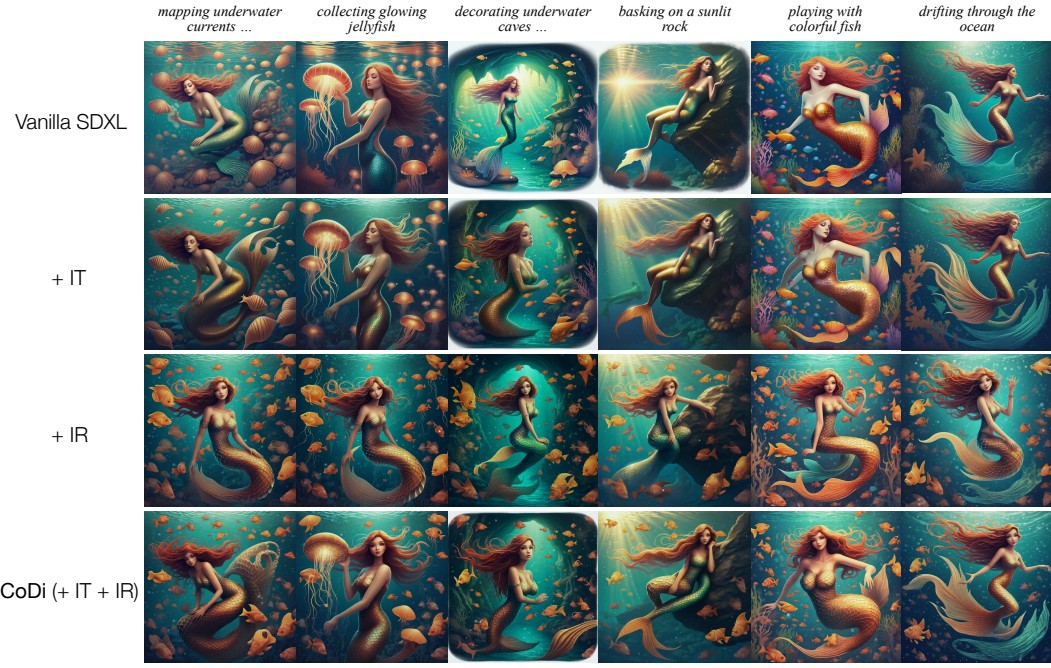

A dreamy underwater illustration of a beautiful mermaid with a shimmering tail.

Figure 4: **Main component analysis (qualitative)** on identity transport (`IT`) and identity refinement (`IR`). `IT` enhances subject consistency in the *coarse-grained level* and preserves pose diversity. `IR` enhances subject consistency in the *fine-grained level* reduces pose diversity. Their combination yields the best consistency and preserves diversity.

Table 2: **Main component analysis (quantitative)** on identity transport (`IT`) and identity refinement (`IR`). `IT` enhances subject consistency and preserves pose diversity. `IR` enhances subject consistency while reduces pose diversity. Their combination yields the best consistency and preserves diversity.

| IT | IR | Subject Consistency | | | Pose Diversity (↑) | Prompt Fidelity (↑) |
|----|----|---------------------|---|---|--------------------|---------------------|
| | | CLIP-I (↑) | DINO-v2 (↑) | DreamSim (↓) | | |
| | | 0.8417 | 0.8010 | 0.3139 | 0.0772 | 0.9082 |
| ✓ | | 0.8576 | 0.8207 | 0.2707 | 0.0800 | 0.9090 |
| | ✓ | 0.8859 | 0.8618 | 0.2044 | 0.0675 | 0.8975 |
| ✓ | ✓ | 0.8809 | 0.8514 | 0.2136 | 0.0758 | 0.9041 |

second only to ConsiStory and comparable to Vanilla SDXL. These results demonstrate `CoDi`'s ability to achieve subject consistency without compromising pose diversity or prompt alignment.

## 4.3 ABLATION STUDIES

**Main component analysis.** The contribution of each module (`IT` and `IR`) to subject consistency and pose diversity are evaluated through quantitative and qualitative ablations, as shown in Table 2 and Figure 4. Table 2 shows that both `IT` and `IR` improve subject consistency, while `IT` also enhances pose diversity. However, using `IR` alone reduces pose diversity—for example, the score drops from 0.0772 to 0.0675 compared to the SDXL baseline. When both modules are applied, subject consistency further improves due to their synergistic effect, while pose diversity is preserved.

Figure 4 visualizes the effect of each module. Compared to Vanilla SDXL, applying `IT` preserves the original pose and improves subject consistency, but some details, such as facial identity, remain suboptimal. In contrast, `IR` alone enhances fine-grained consistency, but produces nearly identical poses across images, substantially reducing diversity. As shown in the bottom row, combining `IT` and `IR` improves both coarse and fine-grained consistency without compromising pose diversity.

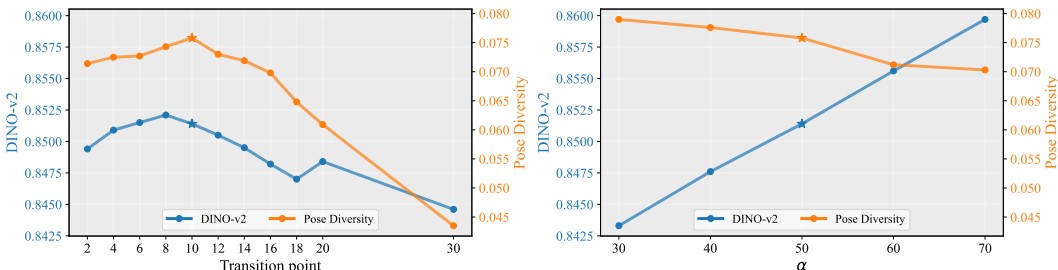

Figure 5: Ablation studies on **(a)** stage transition point, and **(b)** the effect of $\alpha$.

A magical winter illustration with contrasting colors of a brave girl and her fox companion wandering the snow, the fox leads her through the shadows of the trees; a quiet stillness blankets the frozen forest; the sun rises slowly over the frosted hills; the girl rests against the trunk of a frosty pine; the snow catches the pale morning light; the girl tosses a stick into the snow.

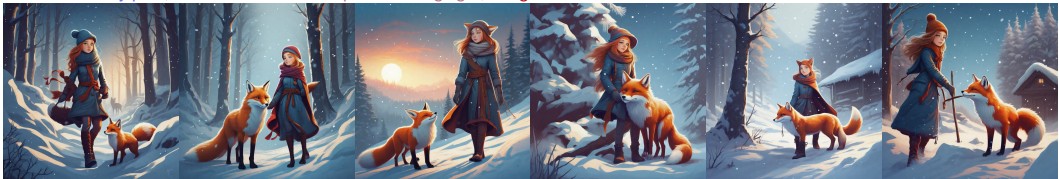

Figure 6: Multi-subject generation. `CoDi` consistently preserves identities of multiple subjects while maintaining diverse poses and spatial layouts.

**Study on stage transition point.** Our `CoDi` adopts a two-stage strategy: identity transport (`IT`) in the early denoising steps and identity refinement (`IR`) in the later ones. By default, we set the stage transition point at step $t = 10$ out of a total of 50 denoising steps (`IT` is applied when $t \leq 10$, and `IR` afterward). In this study, we investigate how the choice of transition point affects generation quality. As shown in Figure 5 (a), we vary $t$ from 2 to 30 and evaluate subject consistency (DINO-v2) and pose diversity. We find that our default choice $t = 10$ achieves a favorable trade-off between consistency and diversity.

**Effect of $\alpha$.** In the `IR` stage, we select the top-$\alpha$ percent of reference features to inject into the target subject features. In this study, we examine how varying $\alpha$ affects subject-consistent generation. Specifically, we vary $\alpha$ from 30% to 70% and report subject consistency (DINO-v2) and pose diversity in Figure 5 (b). We observe that increasing $\alpha$ improves subject consistency but reduces pose diversity. Setting $\alpha = 50\%$ provides a favorable trade-off.

### 4.4 MULTI-SUBJECT GENERATION

Both `IT` and `IR` are independently applied to each subject and are easily extendable, enabling our `CoDi` to naturally support multi-subject generation. For each subject, we perform `IT` and extract its most salient features for `IR`, which effectively prevents feature interference across subjects and enhances subject consistency. As shown in Fig. 6, our `CoDi` preserves multi-subject consistency while maintaining their pose diversity.

### 4.5 DIFFERENT STYLE GENERATION

`CoDi` first transports the identity features from the reference image during the IT stage, and in the IR stage, the diffusion model refines the subject with a specific style. As shown in Fig. 7, CoDi generates images with consistent subject appearance and diverse styles.

### 4.6 USER STUDY

We conducted a user study to compare our method with state-of-the-art approaches. A total of 30 prompt sets were randomly sampled, each consisting of four fixed-length prompts. Thirty-nine participants were asked to evaluate which method demonstrated the best overall performance of the generated images in terms of subject consistency, pose/layout diversity, and prompt fidelity. As

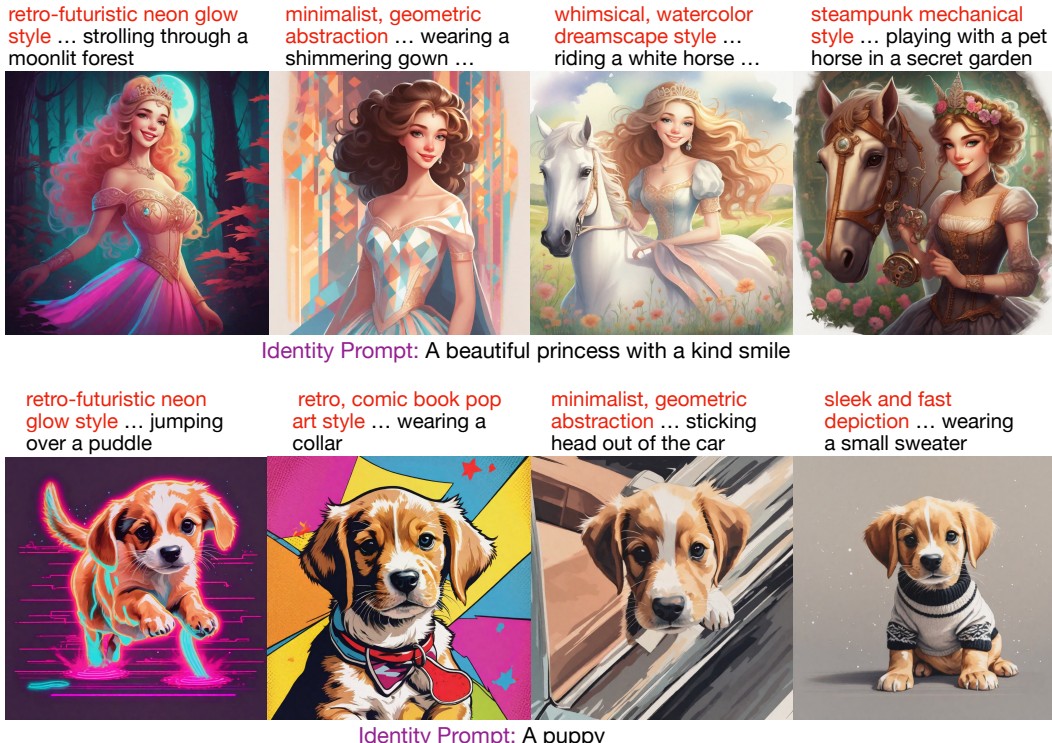

Figure 7: CoDi can generate images with consistent subject appearance and different style.

Table 3: User study with 39 participants evaluating T2I SCG methods based on human preference.

| Method | ConsiStory | StoryDiffusion | 1Prompt1Story | CoDi (ours) |
|---|---|---|---|---|
| Percent (%) ↑ | 19.06 | 21.20 | 14.53 | 45.21 |

shown in Table 3, CoDi achieved the highest overall preference, surpassing the second-best method (StoryDiffusion) by 24.01%.

## 5 CONCLUSION

In this paper, we propose CoDi, a novel training-free framework that addresses the trade-off between subject consistency and pose diversity. CoDi comprises two key components: identity transport (IT) and identity refinement (IR). During early denoising steps, IT aligns features across instances by optimally transporting the identity subject's features, while preserving pose diversity. IR further refines subject consistency by aligning instance features with the salient attributes of the identity subject in the later denoising steps. The effectiveness of our CoDi is demonstrated by its state-of-the-art performance in achieving subject consistency and maintaining pose diversity.

## ACKNOWLEDGMENTS

This work was supported by the Gusu Innovation and Entrepreneur Leading Talents: No. ZXL2024362, Natural Science Foundation of Jiangsu Province: BK20241198, and Natural Science Foundation of China: No. 62406135.

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

**APPENDIX.** This appendix presents supplementary materials that extend the methodological details, experimental evaluations, and analytical discussions introduced in the main body of the paper.

## A  ADDITIONAL IMPLEMENTATION DETAILS

**Extracting subject masks.** We extract subject masks ($M_{\mathsf{id}}$ and $M_n$) by averaging the image-text cross-attention maps over all layers at the final denoising timestep, focusing specifically on subject-related tokens. Let $Q_{\mathsf{img}}$ denote the keys of image features and $K_{\mathsf{sub}}$ the keys of the subject-related tokens. For each cross-attention layer $l$, the unnormalized attention weights are computed as:

$$W_l = \frac{Q_{\mathsf{img}} K_{\mathsf{sub}}^{\top}}{\sqrt{d}}, \tag{11}$$

where $d$ is the feature dimension. We then average the attention weights across all $L$ layers:

$$W = \frac{1}{L} \sum_{l=1}^{L} W_l. \tag{12}$$

We apply Otsu's thresholding (Otsu et al., 1975) to obtain the binary subject mask $M$:

$$M = \mathrm{Otsu}(W). \tag{13}$$

**Selection of the most salient identity features.** Our `IR` refines target images using the most salient reference features, which are determined by the OT plan. Specifically, the saliency score of the $i$-th identity feature is computed as:

$$s_i^{\mathsf{OT}} = \sum_{n=1}^{N} \langle T_n(i,:),\ 1 - C(i,:) \rangle, \tag{14}$$

The top-$\alpha$ index set $\mathcal{I}_i$ in Eq. 9 contains indices with the $\alpha$ highest saliency scores.

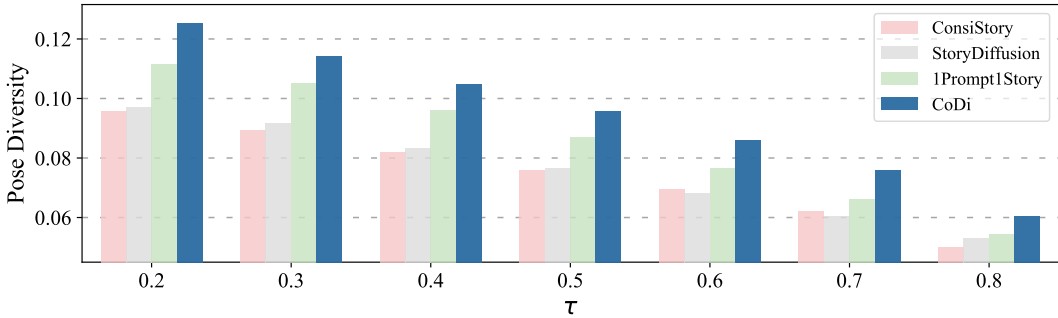

Figure 8: Pose diversity scores across different confidence thresholds $\tau$. Our `CoDi` consistently outperforms other SCG methods by a clear margin under all $\tau$ settings.

## B  ADDITIONAL EVALUATION DETAILS

**Unified evaluation protocol.** We adopt a unified evaluation protocol across all metrics. Specifically, for each target image set $k$ with $N$ generated images $\{\mathbf{x}_n\}_{n=1}^{N}$, we compute the average pairwise evaluation score as follows:

$$u_k = \frac{2}{N(N-1)} \sum_{n=1}^{N-1} \sum_{j=n+1}^{N} f(\mathbf{x}_n, \mathbf{x}_j), \tag{15}$$

where $f(\cdot, \cdot)$ denotes the metric-specific similarity or distance function between two images, depending on the evaluation objective. The final evaluation score is then obtained by averaging $u_k$ over all

target image sets:[2]

$$u = \frac{1}{K} \sum_{k=1}^{K} u_k. \tag{16}$$

**Pose diversity score.** We begin by extracting normalized 2D human keypoints and their confidence scores from each target image using ViTPose (Xu et al., 2022), a SoTA transformer-based model known for its high accuracy and robustness in human pose estimation. Each image $\mathbf{x}$ is represented by a set of $H$ keypoint locations $\mathbf{p}$ and their confidences $\boldsymbol{\beta}$.

$$\mathbf{p} = [(p_1^x, p_1^y)^\top, \ldots, (p_K^x, p_K^y)^\top]^\top \in \mathbb{R}^{H \times 2}, \quad \boldsymbol{\beta} = [\beta_1, \ldots, \beta_K]^\top \in \mathbb{R}^K \tag{17}$$

where each keypoint $\mathbf{p}_i = (p_i^x, p_i^y)$ is normalized by the image width and height and $\beta_i \in [0,1]$ denotes its confidence score. To ensure robustness, we discard keypoints with confidence scores below a threshold $\tau$. For a pair of target images $\mathbf{x}_i$ and $\mathbf{x}_j$, we retain only the indices of keypoints that are valid in both images. We then perform Procrustes method (Schönemann, 1966) to remove global variations in translation, rotation, and scale by aligning $\mathbf{p}_i$ to $\mathbf{p}_j$. Specifically, we first compute the centroids of the keypoints which are denoted as $\boldsymbol{\mu}_i$ and $\boldsymbol{\mu}_j$. We then center both keypoint sets by subtracting their respective centroids and normalize their $\ell_2$ norm:

$$\bar{\mathbf{p}}_i = \frac{\mathbf{p}_i - \boldsymbol{\mu}_i}{\|\mathbf{p}_i - \boldsymbol{\mu}_i\|_2}, \quad \bar{\mathbf{p}}_j = \frac{\mathbf{p}_j - \boldsymbol{\mu}_j}{\|\mathbf{p}_j - \boldsymbol{\mu}_j\|_2} \tag{18}$$

Next, we compute the optimal rotation matrix using singular value decomposition (SVD):

$$\mathbf{U}, \boldsymbol{\Sigma}, \mathbf{V}^\top = \text{SVD}(\bar{\mathbf{p}}_i^\top \bar{\mathbf{p}}_j), \quad \mathbf{R} = \mathbf{V}^\top \mathbf{U}^\top. \tag{19}$$

The resulting $\mathbf{R}$ is an orthogonal rotation matrix that minimizes the Frobenius norm between the aligned keypoint sets, ensuring the best rigid alignment in the least-squares sense. The optimal scaling factor is given by:

$$\gamma = \frac{\|\bar{\mathbf{p}}_j\|_2}{\|\bar{\mathbf{p}}_i\|_2} \cdot \text{tr}(\boldsymbol{\Sigma}). \tag{20}$$

The aligned keypoints are then obtained by applying the computed scale, rotation, and translation:

$$\hat{\mathbf{p}}_i = \gamma \cdot \bar{\mathbf{p}}_i \mathbf{R} + \boldsymbol{\mu}_j. \tag{21}$$

The pose diversity score between a pair of images $\mathbf{x}_i$ and $\mathbf{x}_j$ is computed as the average Euclidean distance between $\hat{\mathbf{p}}_i$ and $\mathbf{p}_j$. To analyze pose diversity under different confidence thresholds $\tau$, we compare the pose diversity scores of various methods across a range of $\tau$ values. As shown in the Fig. 8, our `CoDi` demonstrates a clear advantage over other SCG methods across all $\tau$ settings. We use $\tau = 0.7$ in our experiments to balance keypoint reliability and coverage.

## C LIMITATIONS

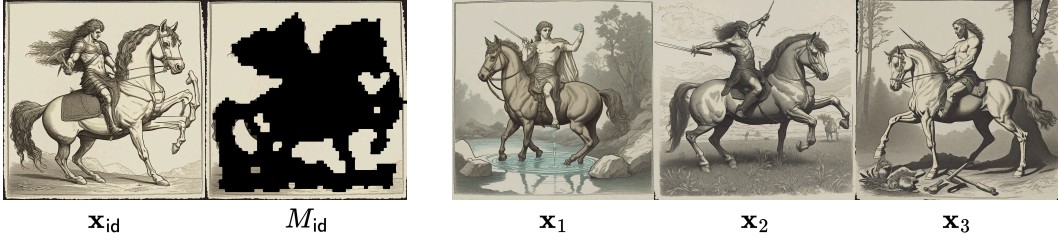

A heroic nature illustration of a centaur with the body of a horse and the torso of a warrior, drinking from a crystal-clear stream; forging weapons in a woodland forge; practicing swordsmanship in a field.

$\mathbf{x}_{\text{id}}$ $\qquad M_{\text{id}}$ $\qquad\qquad\qquad \mathbf{x}_1 \qquad\qquad \mathbf{x}_2 \qquad\qquad \mathbf{x}_3$

Figure 9: Limitations. Our method relies on the quality of cross-attention from the pre-trained diffusion model to accurately localize the subject.

Similar to prior subject-mask-based methods such as ConsiStory (Tewel et al., 2024), our `CoDi` framework relies on cross-attention scores to extract subject masks and estimate image

---

[2]We slightly abuse the notations $K$, which here do not refer to keys in transformer.

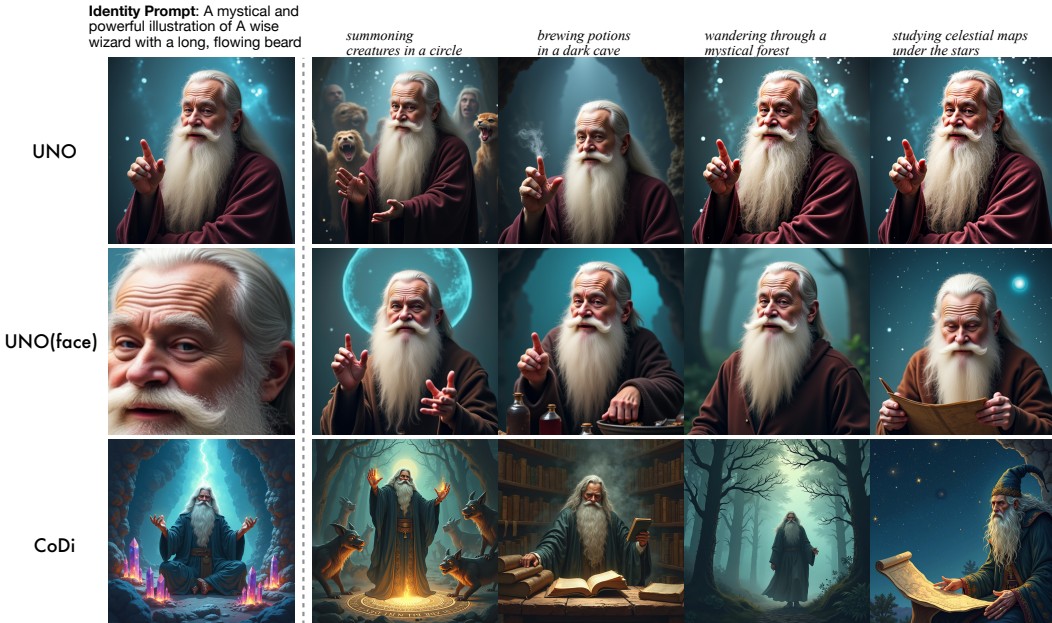

Figure 10: `CoDi` shows superior results compared to SOTA training models.

token importance in the OT Plan. Occasionally, the pre-trained diffusion model assigns higher attention to background regions than to the subject, as shown in Fig. 9, which hinders the effective transport of identity features $\mathbf{X}_{id}$ from the reference image $\mathbf{x}_{id}$ to target image $\mathbf{x}_n$, resulting in subject inconsistency. However, such failures are rare in practice (under 5%) and can be solved by simply changing the seed.

## D    INFERENCE TIME AND MEMORY USAGE

Table 4: **Inference Time and Memory Usage**. We report the inference time (in seconds) and peak GPU memory usage on a single A6000 GPU for generating a set of five images from a prompt set with a resolution of $1024 \times 1024$. StoryDiffusion (Zhou et al., 2024) is excluded due to excessive GPU memory consumption beyond the A6000's limit.

| Method | Inference time (s) | GPU memory (GB) |
|---|---|---|
| Vanilla SDXL (Podell et al., 2023) | 77.67 | 35.58 |
| 1Prompt1Story (Liu et al., 2025) | 115.85 | 17.13 |
| ConsiStory (Tewel et al., 2024) | 113.88 | 46.60 |
| CoDi (ours) | 154.89 | 45.20 |

We measure the inference time and memory usage of different SCG methods on a single A6000 GPU, as shown in Table 4. We report the wall-clock time for generating a set of five images from a prompt set (since the baseline method ConsiStory (Tewel et al., 2024) performs cross-image attention across a batch of images) at a resolution of $1024 \times 1024$. Based on Table 4, our method CoDi exhibits slightly higher inference time (154.89s) and comparable GPU memory usage (45.20GB) relative to ConsiStory. While 1Prompt1Story (Liu et al., 2025) is the most memory-efficient, it compromises subject consistency. Note that we exclude StoryDiffusion (Zhou et al., 2024) due to excessive GPU memory usage beyond the A6000's limit at $1024 \times 1024$ resolution (its original setting uses $768 \times 768$).

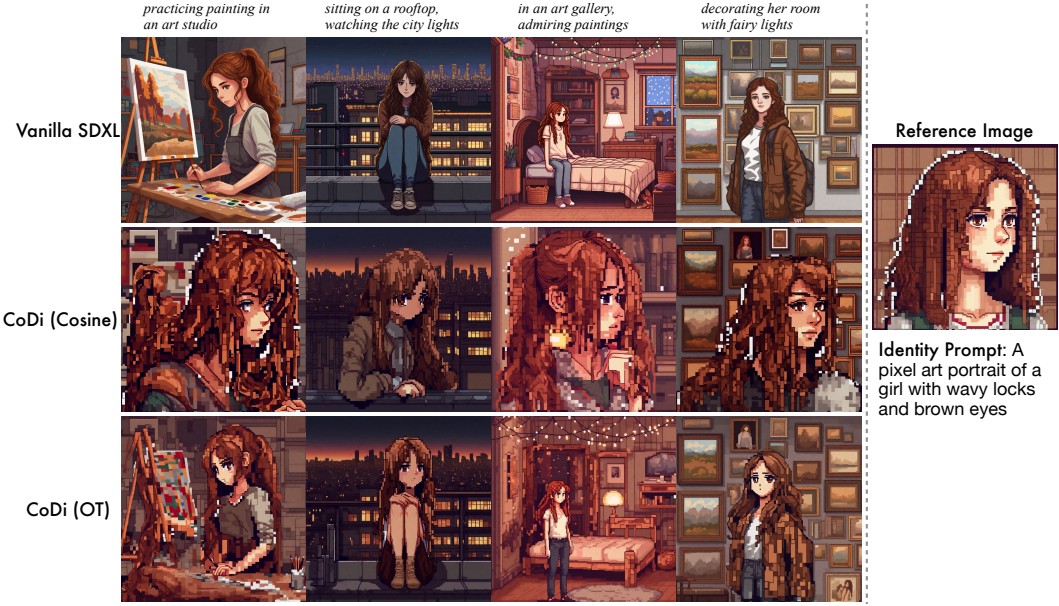

Figure 11: CoDi (Cosine) fails to transport the corresponding features when there is a significant difference between the reference and target images. In contrast, CoDi (OT) achieves overall structural alignment, thereby preserving pose diversity.

## E    COMPARISON WITH SOTA TRAINING METHOD

We compare `CoDi` with UNO (Wu et al., 2025), as shown in Fig. 10. While UNO achieves subject consistency, its identical layout across outputs shows poor action prompt adherence and aesthetics. In contrast, `CoDi` demonstrates improved performance, highlighting the effectiveness of our training-free approach, which shows competitive results compared to SOTA company-level models.

## F    THE EFFECTIVENESS OF OPTIMAL TRANSPORT

To assess the effectiveness of OT, we compare it with a simpler cosine similarity approach. CoDi (OT) achieves 0.0758, significantly outperforming CoDi (Cosine) at 0.0704 in pose diversity. As shown in Fig. 11, when there is a large feature difference between the reference and target images, especially across styles, CoDi (Cosine) tends to transport the mismatched features, failing to preserve the reference image's pose. In contrast, CoDi (OT) effectively maintains the pose while enabling diverse variations, resulting in a failure to preserve the reference image's pose. In contrast, CoDi (OT) effectively preserves the pose of the reference image while maintaining diverse pose variations.

## G    GENERALIZATION TO DiT-BASED ARCHITECTURES

We adapt `CoDi` to DiT-based models (Flux), with the generation results shown in Fig. 13, demonstrating richer details, subject consistency, and pose diversity.

## H    USER STUDY DETAILS

We conducted a user study comparing our method with state-of-the-art approaches, including ConsiStory, StoryDiffusion, and 1Prompt1Story. Thirty prompt sets, each containing four fixed-length prompts, were randomly sampled. Thirty-nine participants evaluated which method achieved the best overall image quality in terms of subject consistency, pose/layout diversity, and prompt fidelity.

Participants were instructed to select the set that best satisfied three evaluation criteria: subject consistency, pose/layout diversity, and prompt fidelity. Fig. 14 illustrates these criteria at the start

*1. walking down the street*    *2. talking with a passerby on the corner*    **Identity Prompt:** A fashion illustration style of a short-haired man wearing a black suit and a red tie

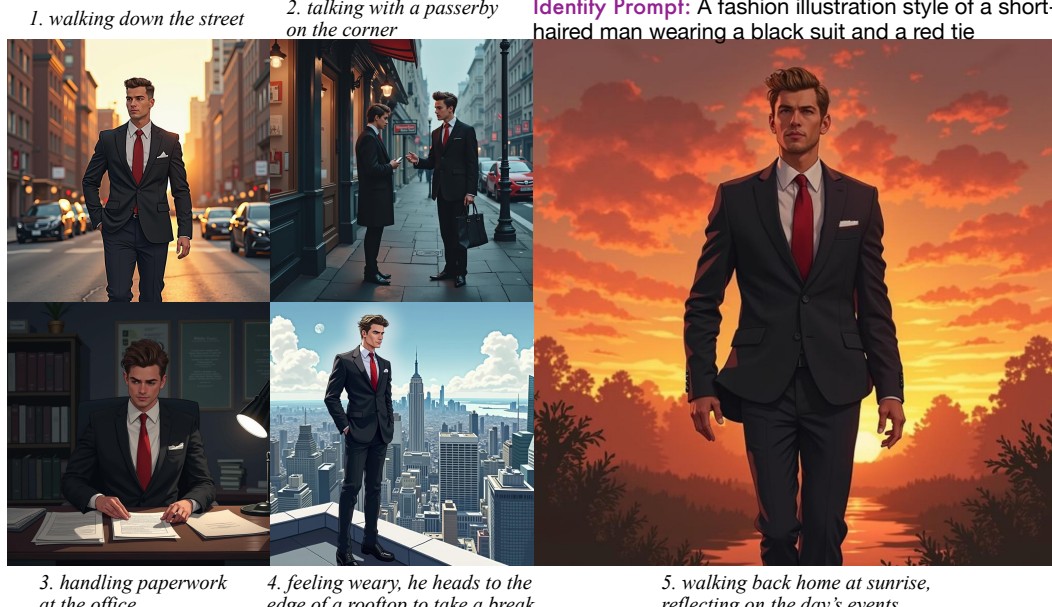

*3. handling paperwork at the office*    *4. feeling weary, he heads to the edge of a rooftop to take a break*    *5. walking back home at sunrise, reflecting on the day's events*

Figure 12: `CoDi` can maintain both subject and clothing consistency while preserving pose diversity by adding the clothing description in the prompt and adjusting the subject mask threshold to include clothing in the foreground mask.

A sleek and fast depiction of a cheetah with sharp eyes, sprinting across the savannah; stalking a gazelle in the grass; relaxing in the shade under a tree; marking territory with its scent.

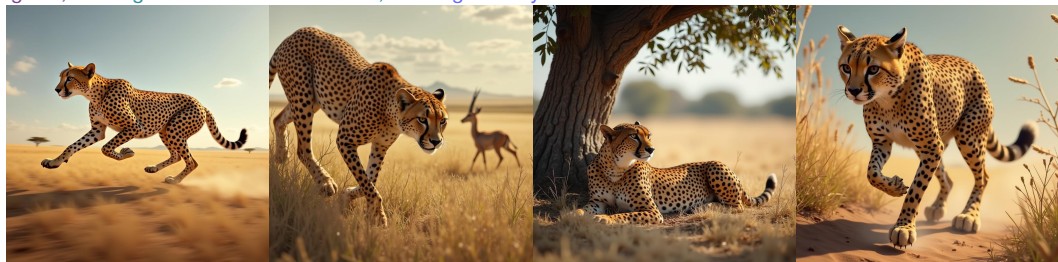

A 3D animation of a black and white dog with yellow collar, wearing a bandana; biting a bone; wearing a birthday hat; sitting by a fireplace.

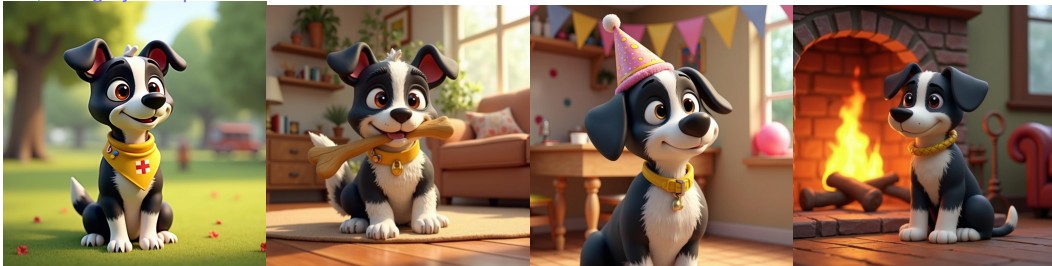

Figure 13: We have extended `CoDi` to the DiT-based architecture (Flux), with the generated results maintaining subject consistency while achieving diverse pose diversity.

of the questionnaire. To facilitate informed selections, a representative example was provided, accompanied by a performance comparison and rationale, highlighting the reasoning behind choosing the optimal set.

# I  ADDITIONAL RESULTS

We present additional qualitative comparisons in Fig. 15, along with more results generated by our `CoDi` in Fig. 16. These examples further demonstrate that our method achieves state-of-the-art performance in subject consistency, pose diversity, and prompt fidelity. In contrast, existing SCG methods remain limited, often excelling in only one or two of these aspects—typically at the expense of pose diversity or subject consistency.

**Long story generation.** As each target image $\mathbf{x}_n$ relies solely on reference image $\mathbf{x}_{\mathrm{id}}$ for subject identity, our `CoDi` enables extended visual storytelling. As demonstrated in Fig. 17, it maintains subject consistency across diverse prompt semantics, supporting the generation of varied layouts and poses. This makes `CoDi` effective for long-form generation, where both prompt fidelity and visual diversity are essential.

# J  THE USE OF LARGE LANGUAGE MODELS (LLMs)

In our work, large language models (LLMs) were employed primarily for general writing assistance. Specifically, we used LLMs to refine sentence expressions, check for grammatical errors, and convert tables into LaTeX format.

## Evaluation Criteria

The questionnaire includes 30 questions, each showing four image sets generated by different methods. For each question, please choose the set that best meets the following criteria (evaluated within each row of images):

1. **Subject Consistency**: the subject should maintain a consistent appearance across the row.
2. **Pose/layout Diversity**: the subject should exhibit varied poses and spatial arrangements across the row.
3. **Prompt Fidelity**: Alignment with given text descriptions.

## Example

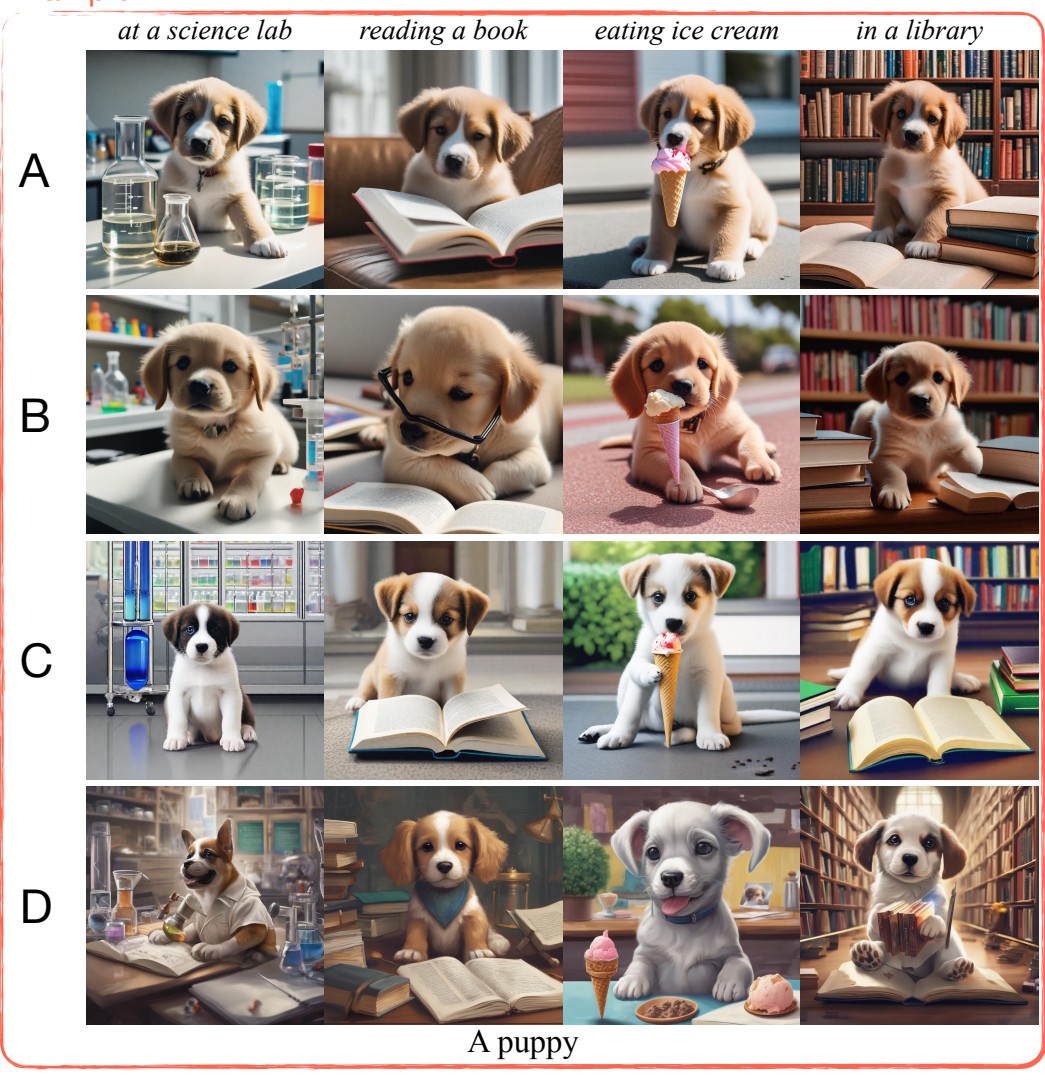

A puppy

## Performance Comparison and Selection Rationale

Row A: Although it achieves subject consistency, the poses and layouts are nearly identical.
Row B: Performs well across subject consistency, pose/layout diversity, and prompt fidelity.
Row C: Shows poor subject consistency.
Row D: Shows poor subject consistency.

**Overall, Row B represents the best choice.**

Figure 14: **Questionnaire of user study.** Evaluation Criteria outlines the standards for selecting image sets. Example illustrates a representative question for demonstration purposes. Performance Comparison and Selection Rationale demonstrates how the best choice is determined based on the example. These sections visually convey the evaluation criteria and guide participants' selections.

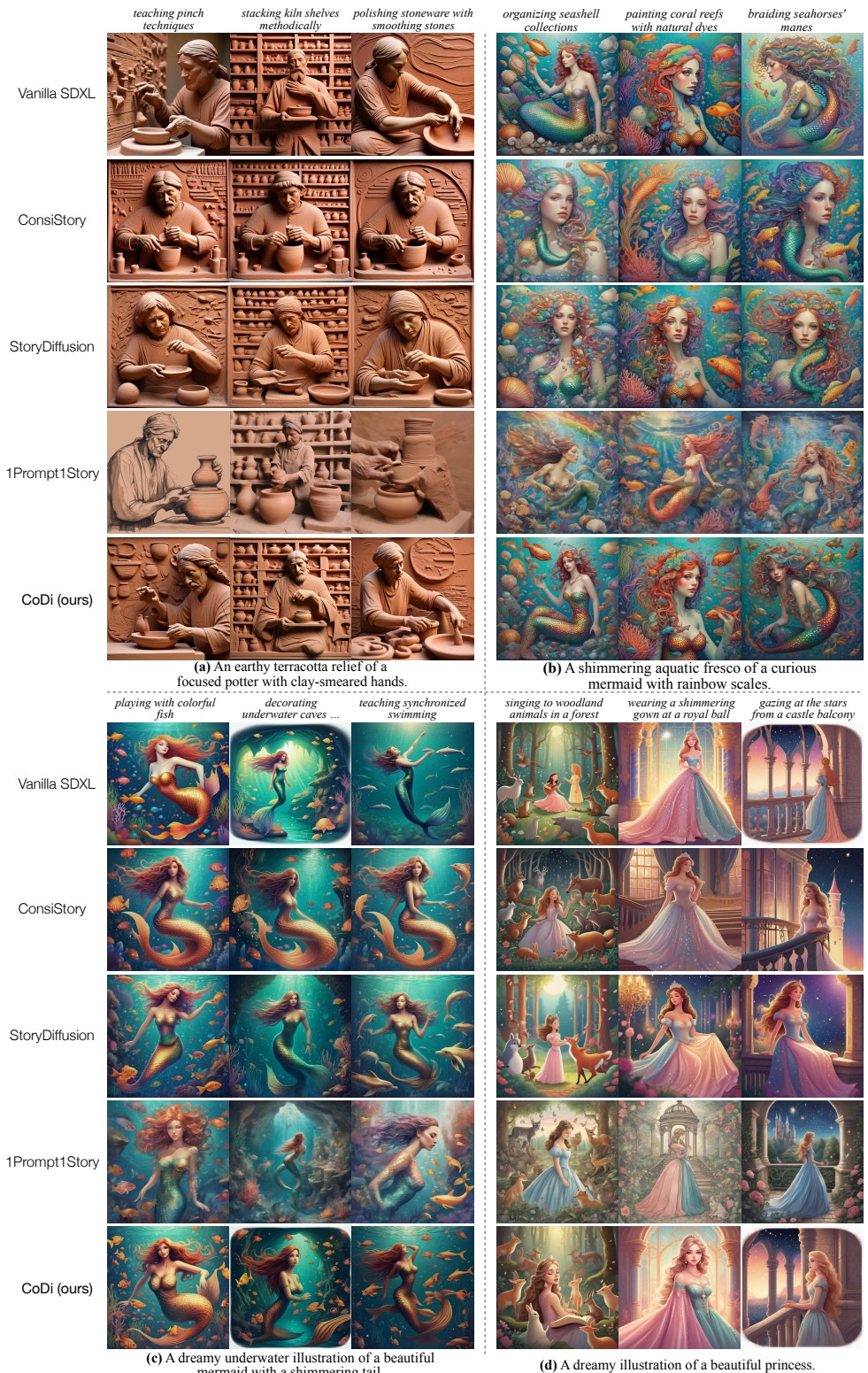

Figure 15: Additional qualitative comparisons. Our `CoDi` achieves the best trade-off among subject consistency, pose diversity, and prompt fidelity.

A pastoral countryside painting of a flour-dusted miller in apron, stacking flour sacks; sifting golden grains; reading wind signs; restoring waterwheel mechanisms; testing flour quality; bagging fresh products.

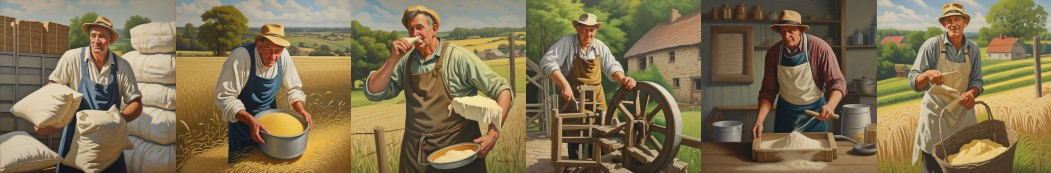

A haunting painting of a seductive siren, weaving sea fog; resting on a sunken ship's mast; whispering to dolphins in the moonlight; weaving seaweed into magical charms; floating among glowing jellyfish in the deep sea; playing an enchanted harp on a rocky shore.

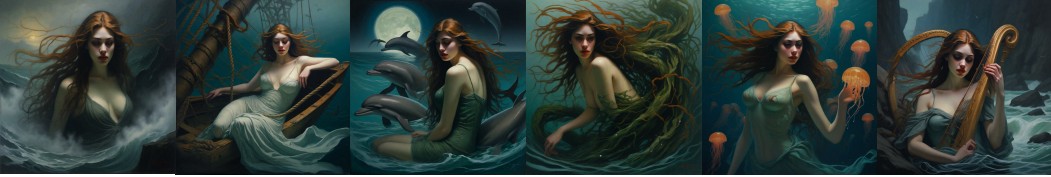

A peaceful and rustic illustration of a hobbit with large feet and a warm smile, enjoying a feast at home; lounging by a cozy fireplace; walking through the rolling hills; baking bread in a stone oven; reading a book in a sunlit nook; harvesting vegetables from the field.

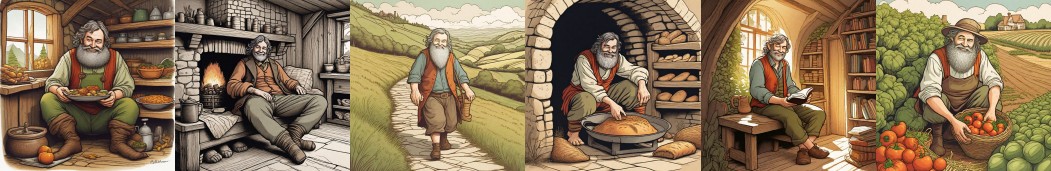

A terrifying and wild illustration of a werewolf with glowing yellow eyes, snarling beside a flickering campfire; resting with a pack under moonlight; glaring through the cracks of a cabin wall; tracking footprints through fresh snow; standing atop a hill under a blood moon; crouching behind tall reeds by a lake.

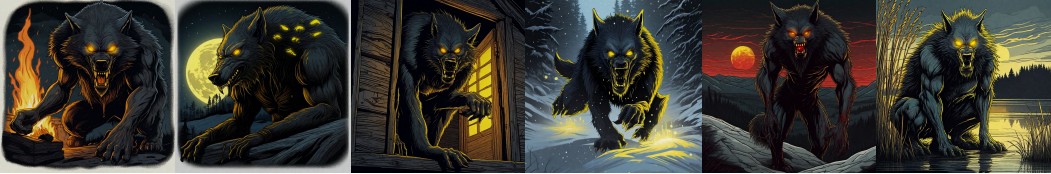

A graceful and elegant illustration of a beautiful princess in a flowing gown, befriending a dragon atop a cliff; contemplating her destiny in a candlelit chapel; feeding swans by the lake; reading in the palace library; walking through a royal garden; lighting lanterns during a peace ceremony.

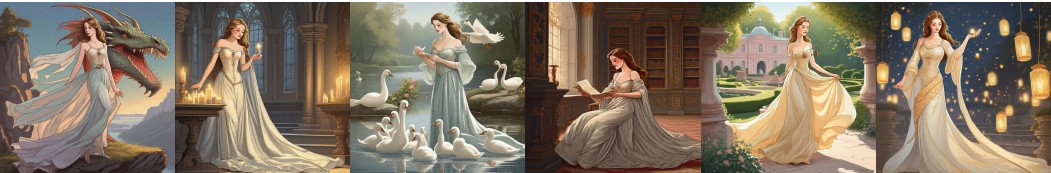

A 3D animation of a happy hedgehog, dressed in a festive outfit; hiding inside a boot; in a cozy nest; dressed in a miniature jacket; nibbling on a strawberry; wearing round glasses.

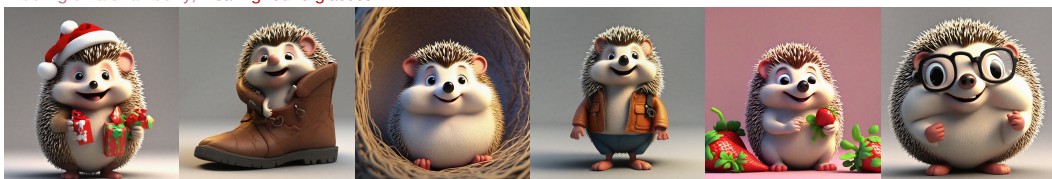

Figure 16: Additional qualitative results generated by our CoDi demonstrate strong subject consistency and pose diversity.

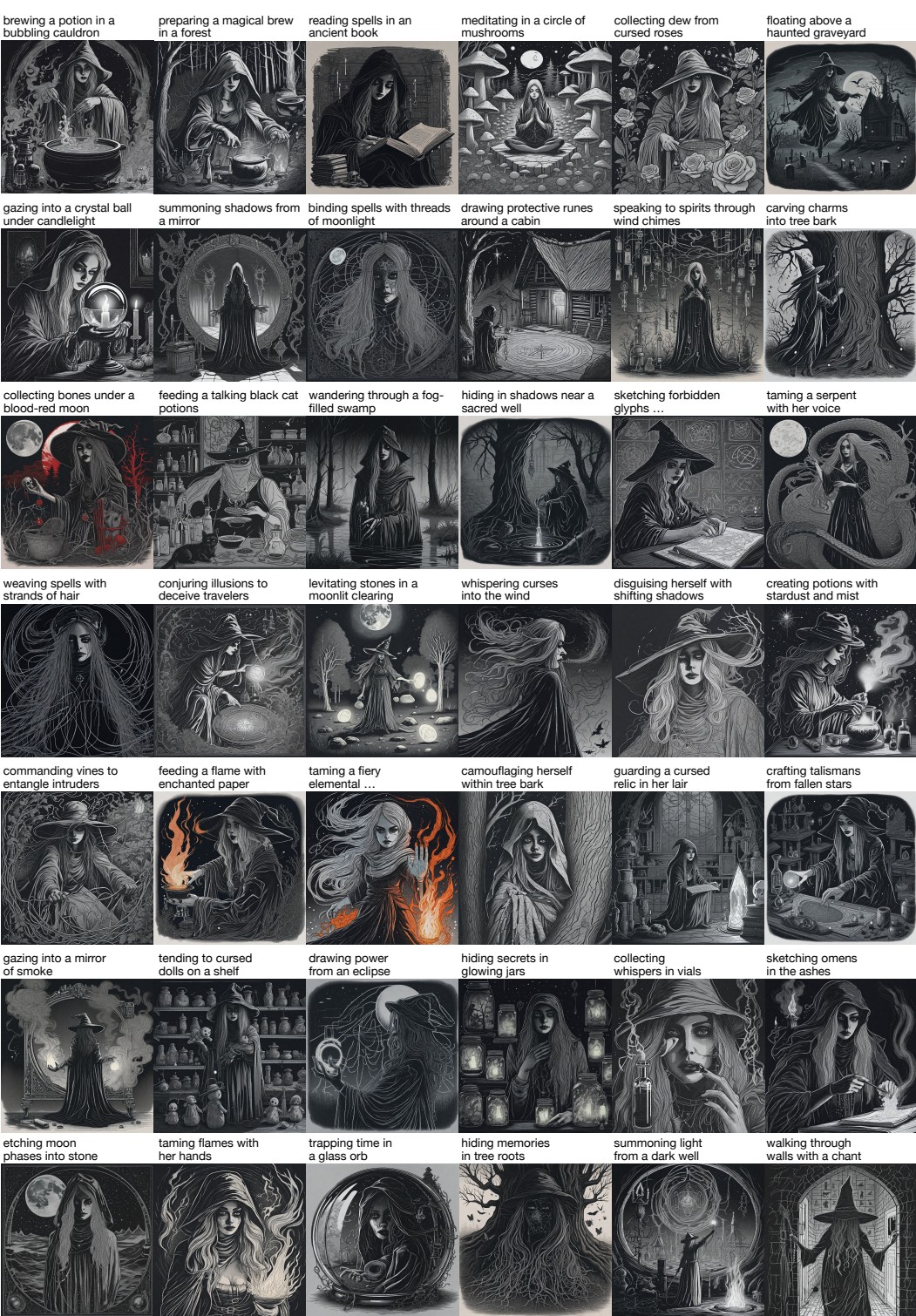

Identity Prompt: A dark drawing of a mysterious witch.

Figure 17: Long Story Generation. CoDi supports extended visual storytelling by generating diverse scene compositions while consistently preserving subject identity throughout the sequence.

