# OpenReview forum: "CoDi: Subject-Consistent and Pose-Diverse Text-to-Image Generation"
_ICLR.cc/2026/Conference — ICLR 2026 Poster_

### Official Review · Reviewer_VX3M · 2025-10-27

**Soundness:** 3
**Presentation:** 3
**Contribution:** 1
**Rating:** 2
**Confidence:** 5

**Summary:**

This work focuses on subject-consistent generation. It preserves pose diversity by leveraging Identity Transport in the early stage of denoising and promotes subject consistency through Identity Refinement in the late stage of denoising. Both qualitative and quantitative experiments demonstrate the effectiveness of the proposed method.

**Strengths:**

- The paper is well written and easy to follow.
- The method proposed in the paper is training-free and can be directly applied during inference.
- The paper includes comprehensive comparative experiments and ablation studies. The design of evaluation metrics also demonstrates certain insights, especially regarding "pose diversity".

**Weaknesses:**

- Optimal Transport (OT) essentially addresses the optimization problem of transforming one probability distribution into another with minimal cost. However, the problem in **IT** is to find the feature matching relationship between the reference image and the target image. Clearly, a straightforward ranking based on cosine similarity would be simpler and more efficient. In contrast, the "globally optimal transport" property of OT not only complicates the problem but may also introduce redundancies. To illustrate this with a simple though imperfect analogy: The IT task aims for direct and semantically consistent matches. For instance, it seeks to align the "eye" feature in the reference image with the "eye" feature in the target image, and the "nose" feature with the "nose" feature. Introducing OT could disrupt this consistency. Suppose the "eye" feature in the reference image has a higher correlation than other features. Under OT’s global optimization logic, this could lead to the "eye" feature being inappropriately "transported" to the "nose" region in the target image. This outcome violates the need for direct and semantically meaningful feature matching. In summary, I argue that the introduction of Optimal Transport in this work serves more as a mathematical formality rather than a practically useful component.
- The design of Identity Refinement lacks innovation, as its feature fusion approach bears significant similarities to that of prior works.
- SDXL is built on the U-Net architecture, which is considered outdated in the current field. Most state-of-the-art base models now adopt the DiT architecture, and the effectiveness of the proposed method when applied to DiT-based models remains unproven.

**Questions:**

Given that current state-of-the-art base models such as XVerse, UNO, Flux-Kontext and Nano Banana have already achieved the three key objectives outlined in the introduction with excellent performance, and further support unlimited multi-subject consistent generation as well as strong generalization capabilities, what is the core significance of this work?

---

> ### Author Response · Authors · 2025-11-22
>
> We appreciate your insightful comments and detailed remarks.
>
> **W1: The introduction of Optimal Transport in this work does not provide a practically useful component.**
>
> **A1:** **Response to the concern about using Optimal Transport (OT) instead of cosine similarity**
> - **1) Why OT rather than cosine similarity — empirical justification**
>
> In the early development of this work, we conducted experiments comparing cosine similarity with Optimal Transport (OT) during the IT stage. While both approaches exhibit similar performance in terms of subject consistency and prompt fidelity, OT produces **substantially superior pose diversity**. For example, OT improves the pose diversity metric by **0.0096** over 1Prompt1Story, representing a **2.29× increase** compared to the **0.0042** improvement achieved by cosine similarity. As illustrated in Fig.8 of the revised paper, when there exists a significant feature discrepancy between the reference and target images (especially across different artistic styles), cosine similarity often *misaligns* the features of less prominent regions, causing the resulting image to deviate from the intended reference pose. OT, on the other hand, consistently preserves the pose structure while enabling diverse pose variations—an aspect our method prioritizes. Because pose diversity is central to the objective of our approach, we therefore adopt OT.
> - **2) Computational efficiency**
>
> Regarding computation, OT imposes negligible overhead in practice. The matching between the reference and target features is computed only once per shot, and the observed inference-time difference between OT and cosine similarity is typically **well under one second**. Thus, OT does not introduce a meaningful computational burden.
> - **3) The reviewer’s conceptual example actually supports our use of OT**
>
> We appreciate your illustrative example. Interestingly, it highlights precisely **the mode that cosine similarity fails but OT can mitigate**. When the “eye” feature in the reference image happens to have a high cosine similarity with the “nose” region of the target image, cosine similarity assigns matches independently and greedily, leading to incorrect and semantically implausible correspondences (e.g., eye → nose). This behavior is unavoidable because cosine similarity considers each feature in isolation. In contrast, OT optimizes *global* consistency by minimizing the total transport cost across all correspondences. Even if the “eye” feature has slightly higher affinity to the “nose” region, OT will still map “nose → nose” and “eye → eye” because doing so achieves a lower *overall* assignment cost. As a result, OT preserves structural alignment and pose coherence—exactly the requirement of the IT stage.
>
> | Method        | Pose-Diversity ↑ |
> | :------------ | ---------------: |
> | CoDi (Cosine) |           0.0704 |
> | CoDi (OT)     |           0.0758 |
>
> **W2: The design of Identity Refinement lacks innovation, as its feature fusion approach bears significant similarities to that of prior works.**
>
> **A2:** Identity Refinement (IR) is not the key innovation of this paper. The main contribution is the proposal of a two-stage strategy that decouples pose diversity and subject consistency, with the IT stage addressing pose diversity by matching feature distributions between the reference and target images. While IR resembles cross-image attention-based SCG methods, it introduces two distinctions: **1)** each target image attends only to the reference image, avoiding entangled feature updates across target images **2)** each target image attends only to the most relevant features from the reference image. These simple yet effective design choices enhance the scalability for long-story generation and effectively preserve pose diversity, as demonstrated in the ablation study.
>
> **W3: Is the proposed method still effective when applied to DiT-based models?**
>
> **A3:** Yes, We successfully adapted CoDi to DiT-based models (Flux), with generation results in Fig.9 of the revised paper showcasing richer details, subject consistency, and pose diversity.
>
> **Q1: What is the core significance of this work compared to state-of-the-art models like XVerse, UNO, Flux-Kontext, and Nano Banana?**
>
> **A4:** Unlike these SOTA models, which rely on extensive data and computational resources, CoDi is a **training-free** approach that eliminates parameter tuning, ensuring strong generalization and broad compatibility across various diffusion architectures. As shown in Fig.7 of the revised paper, we compare our results with UNO. While UNO achieves subject consistency, its identical layout across outputs fails to adequately follow action prompts and lacks aesthetic variation. In contrast, CoDi demonstrates improved performance, highlighting the effectiveness of our training-free approach, which delivers competitive results relative to SOTA training models. The discussion is included in the related work section of the revised paper.

---

> ### Comment · Reviewer_VX3M · 2025-11-24
>
> #### Response to A1：
>
> > 3\) The reviewer’s conceptual example actually supports our use of OT
>
> In my review, when I mentioned "higher correlation," I was explicitly referring to the definition of **probability mass** as described in Line 238 of the paper (where feature relevant to the entire subject are used as mass), *not* the cosine similarity between feature vectors as the authors assumed in their rebuttal.
>
> My concern remains valid: If the "eye" feature in the reference image possesses significantly higher probability mass (importance) than the corresponding region in the target image, **standard OT constraints will force this excess mass to be transported to semantically unrelated regions (e.g., the "nose") to satisfy global marginal constraints**. This is a fundamental limitation of OT in unbalanced scenarios, which violates the requirement for semantic consistency. The authors' rebuttal avoids this core issue by pivoting to a scenario of "high cosine similarity between eye and nose," which was not the point of my critique.
>
> I must admit that I was initially persuaded to raise my rating, as the precise context of the paper and my initial critique had slightly faded from memory. However, upon reviewing the specific definition provided in Line 238, I realized the argument was not valid. *This technique of misrepresenting the reviewer's point and misleading the reviewers is highly inappropriate for academic discourse*. By ignoring their own definition of mass, the rebuttal fails to address how their method handles the structural misalignment caused by mass imbalance. **Therefore, I am not convinced.**
>
> > 1\) empirical justification
>
> The authors’ attempt to address a foundational **theoretical concern** with **empirical justification** is conceptually insufficient and fails to satisfy the requirement for scientific rigor.
>
>
>
> #### Response to A2：
>
> Regarding the two claimed distinctions of the IR, these points appear to be relegated to engineering choices rather than fundamental novelties. More critically, the first distinction has been previously demonstrated in existing work; specifically, please refer to ConsiStory, Section 4.4.
>
> Furthermore, while the authors clarify that their main contribution is the two-stage strategy that decouples pose diversity and subject consistency, I contend that this is **at best incremental**. The core methodology—prioritizing the establishment of pose diversity before enforcing stringent subject consistency—is a concept that has been previously explored and established in the literature, please refer to ConsiStory, Section 4.2.
>
> #### Response to A3：
>
> I appreciate this clarification and the inclusion of results based on DiT-based models (Flux). The successful adaptation to state-of-the-art architectures significantly elevates the practical relevance and potential impact of the proposed method.
>
> #### Response to A4：
>
> The main advantage of SOTA methods is their paradigm: they use **one single reference image** to generate **unlimited consistent images**. This is much more flexible and better for generalization than CoDi's **all-at-once** generation approach. Furthermore, your criticism about UNO’s "identical layout" is easily solved in that paradigm by simply using a reference image that only shows the subject's face. Therefore, the question still stands: What is the true significance of this work compared to the more flexible and widely used single-reference SOTA models?
>
> In summary, I maintain my score and hold a firm conviction in this assessment.

---

> > ### Author Response · Authors · 2025-11-25
> >
> > We thank the reviewer for the thoughtful response.
> >
> > **R1: We do not misrepresent the reviewer's point or mislead the reviewers.**
> > > Original review: Suppose the "eye" feature in the reference image has a higher correlation than other features. Under OT’s global optimization logic, this could lead to the "eye" feature being inappropriately "transported" to the "nose" region in the target image.
> >
> > >  Recent response : I was explicitly referring to the definition of **probability mass**
> >
> > The reviewer's original response does not explicitly refer to the definition of "probability mass," and the term "probability mass" does not appear in the original review. The higher correlation mentioned by the reviewer, along with the "eye" and "nose" example, could be more easily interpreted as suggesting that the "eye" has a higher similarity to the "nose" region of the target image than the "nose" in the reference image. Our response carefully considers the meaning of the original review. Therefore, the reviewer's claim that the authors are "misrepresenting the reviewer's point and misleading the reviewers" is inappropriate.
> >
> > **R2: On Local Probability Mass Disparity in Standard OT**
> >
> > We appreciate the reviewer's deep insight into the theoretical constraints of Optimal Transport. The reviewer is theoretically correct that standard OT enforces mass conservation, but the concern that excess mass shifts to specific, incorrect semantic regions (e.g., eye $\to$ nose) overlooks the dominance of the **Cost Matrix $C$** in the optimization objective.
> >
> > **1) Cost Matrix Mechanism & Semantic Preservation**
> > The OT objective, $\underset{T\geq0}{\min}\langle T,C\rangle$, ensures that the transport plan minimizes the total cost. Since the feature dissimilarity between semantically distinct regions (like "eye" and "nose") is high, mapping surplus mass to the 'nose' would incur a **prohibitive cost**. Instead, the optimization algorithm achieves mass balance by:
> > - **Diffusing Excess Mass:** The surplus mass is distributed **diffusely across regions of minimal semantic penalty** (i.e., areas with lower feature conflict), resulting in a **high-entropy distribution**. This dispersal prevents the mass from concentrating on any specific, incorrect semantic target.
> > - **Preserving Structure:** Consequently, the semantic structure is preserved because the **peak transport probability** remains fixed on the correct matching regions, while the 'leakage' is effectively suppressed by the high cost of semantic mismatch.
> >
> > **2) OT as a Soft Attention Mechanism**
> > Crucially, our framework interprets the transport plan $T$ as a **soft attention mechanism** rather than a hard assignment. The eye → eye connection, driven by high feature similarity, retains the **dominant probability weight**—this is the meaningful signal. The "leakage" caused by mass constraints results in negligible weights spread, which act merely as low-magnitude noise.
> >
> > In summary, the erroneous semantic mapping feared by the reviewer does not degrade the feature aggregation performance, as the meaningful signal decisively dominates the noise.
> >
> > **R3: The empirical justification is in response to the reviewer's summary that OT does not provide a practically useful component.**
> >
> > In the original review, the reviewer summarized that OT does not provide a practically useful component.
> > The reviewer also mentioned that cosine similarity would be a better choice.
> > >Clearly, a straightforward ranking based on cosine similarity would be simpler and more efficient
> >
> > In response, we provide empirical justification to demonstrate that OT is practically superior to cosine similarity.
> > > The authors’ attempt to address a foundational **theoretical concern** with **empirical justification** is conceptually insufficient
> >
> > Therefore, our goal is not to address the foundational theoretical concern with empirical justification, but to validate that OT is indeed a practically useful component.

---

> > ### Author Response · Authors · 2025-11-25
> >
> > **R4: Distinctions from ConsiStory**
> >
> > While ConsiStory in Section 4.2 discusses prioritizing pose diversity, it is primarily a mitigating technique within a coupled system. Its query-blending mechanism and attention dropout aim to address overly consistent pose diversity, but they fail to solve the underlying issue, sacrificing subject consistency in the process. In contrast, our approach **decouples** these objectives, allowing us to enforce **maximal diversity** in the first stage and **maximal consistency** in the second. Our experimental results show that CoDi not only outperforms ConsiStory in pose diversity but also surpasses it in subject consistency.
> > Identity Refinement (IR) is not the central innovation of this paper.
> >
> >
> > **R5: The significance of this work compared to single-reference SOTA models.**
> >
> > Although both CoDi and these SOTA models achieve subject consistency, they are different approaches. CoDi is a text-to-image model, while they are text&image-to-image models. Additionally, CoDi can also generate unlimited consistent images using a reference image generated from the identity prompt.
> >
> > Regarding UNO’s "identical layout": The reviewer’s suggested approach of using a reference image that only shows the subject's face is followed. Specifically, the reference image is first generated using the identity prompt, and then the subject's face is cropped to serve as the reference image for UNO (face). As shown in the updated results in Fig. 7, the generated outputs for UNO (face) still suffer from the identical layout issue. Additionally, the results demonstrate poor prompt adherence; for instance, in P1 of UNO (face), no creatures appear. Based on these generated images alone, it is difficult to discern the semantic differences.
> >
> > In summary, the significance of CoDi lies in its training-free approach, which does not require extensive data training or resource-intensive processes, yet still achieves competitive, and in some cases, superior results compared to SOTA models.

---

> > > ### Comment · Reviewer_VX3M · 2025-11-28
> > >
> > > I appreciate the authors' clarification of the misunderstanding. While this work exhibits some conceptual similarity to prior efforts, its novelty remains sufficient for the ICLR bar. Additionally, although I still harbor reservations about the suitability of standard Optimal Transport (OT) for this modeling task, the authors' experiments empirically confirm its effectiveness. The underlying operational mechanism is a compelling subject for future exploration. Based on the consolidated feedback from the other reviewers and the authors' rebuttal, I have decided to raise my score to 6.

---

> > > > ### Comment · Reviewer_VX3M · 2025-11-28
> > > >
> > > > I think I ran into a bug here because the 'Edit' button disappeared, and even switching browsers didn't help. It's likely on the system side. Just give me a heads-up once it's working again, please.

---

> > > > > ### Author Response · Authors · 2025-11-28
> > > > >
> > > > > We sincerely thank you for your valuable feedback and for raising the score to 6. We are glad that our response helped clarify the misunderstandings, and we truly appreciate your recognition of our work's novelty.
> > > > >
> > > > > We also noted your comment regarding the system glitch with the "Edit" button. Thank you for posting the update to keep us informed.

---

### Official Review · Reviewer_rZBz · 2025-10-31

**Soundness:** 3
**Presentation:** 4
**Contribution:** 2
**Rating:** 6
**Confidence:** 4

**Summary:**

To solve the trade-off between subject consistency and pose diversity in text-to-image subject-consistent generation (SCG), the paper proposes the CoDi framework. Inspired by diffusion models’ progressive nature, this paper includes two stages: Identity Transport and Identity Refinement.  Extensive experiments have demonstrated the effectiveness of the model.

**Strengths:**

1. This work proposes an effective method to improve pose diversity.
2. This work is clearly expressed and easy to understand.
3. This work introduces Optimal transport into Subject-consistent generation.

**Weaknesses:**

1. This model was tested on SDXL, but its effectiveness was not verified on the DiT architecture.
2. The qualitative and quantitative experimental results of this work did not show significant improvement.
3. The long description in lines L126-L131 seems informal in the main text.

**Questions:**

1. This work was conducted on the SDXLl, which is a relatively old base model. Could the proposed method in this paper work on DiT model (e.g., FLUX and SD 3.5)?
2. In the experimental results shown in Figure 3(a) of this paper, the styles of the three scientists are not very consistent (and all are anime characters), and the portrait similarity is also lower compared to ConsiStory and StoryDiffusion. Will the proposed model in this paper affect the style?
3. Could this method generate images with consistent subject appearance and different style?

---

> ### Author Response · Authors · 2025-11-22
>
> Thank you for your review and constructive feedbacks.
>
> **W1&Q1: Is the model still effective when applied to the DiT architecture?**
>
> **A1:** Yes, we successfully adapted CoDi to DiT-based models (Flux), with generation results in Fig.9 of the revised paper showcasing richer details, subject consistency, and pose diversity.
>
> **W2: The quantitative and qualitative experimental results of this work did not show significant improvement.**
>
> **A2:**
> - **1)** In the quantitative experiment (Table 1), CoDi achieves the highest pose diversity score, surpassing the second-best model, 1Prompt1Story, by 0.0096, which reflects a **2.34x** improvement over the 0.0041 increase observed for 1Prompt1Story over ConsiStory.  In terms of subject consistency, our method achieves the lowest DreamSim score, improving by **8.56%** over the second-best model, ConsiStory.
> - **2)** For qualitative experiment (Fig.3), CoDi maintains subject consistency while achieving diverse pose diversity. In contrast, ConsiStory and StoryDiffusion achieve subject consistency at the cost of pose diversity. We encourage the reviewer to also refer to the generated results of CoDi in the DiT-based architecture (Fig. 9), which demonstrate subject consistency, pose diversity, and enhanced details and aesthetics.
> - **3)** Furthermore, the user study results in Table 3 show that CoDi achieved the highest overall preference (45.21%), surpassing the second-best method, StoryDiffusion (21.20%), by **24.01%**. This highlights a significant improvement and better alignment with human preferences.
>
> **W3: The long description in lines L126-L131 seems informal in the main text.**
>
> **A3:** Thank you for your valuable suggestion. We have revised the text accordingly.
>
> **Q2: In Fig.3(a), the styles of the three scientists are inconsistent. Does the proposed model affect the style? Why are all the scientists anime characters? Why is the portrait similarity is lower compared to ConsiStory and StoryDiffusion?**
>
> **A4:**
> - **1)** CoDi doesn't affect the style.  The observed style variation likely stems from differences in the backgrounds of our results, as CoDi preserves the background of the original target image (generated by Vanilla SDXL). Based on our experiments and the additional results presented in the paper, CoDi does not alter the style.
> - **2)** CoDi generates each target image based solely on the reference image for subject identity. Since the reference image is an anime character, all three target scientists are all anime characters to maintain consistent identity
> - **3)** The reference image is an anime character, while the target images are realistic (as shown in Vanilla SDXL in Fig. 3(a)), which causes a large feature difference that challenges subject consistency. Nevertheless, CoDi successfully maintains subject consistency while allowing for diverse pose variation. In contrast, ConsiStory and StoryDiffusion operate solely on attention mechanisms across target images, making subject consistency easier to achieve when all target images are of real people. Our experiments show that CoDi results in fewer inconsistent examples than both ConsiStory and StoryDiffusion.
>
> **Q3: Could this method generate images with consistent subject appearance and different style?**
>
> **A5:** Yes, CoDi first transports the identity features from the reference image during the IT stage, and in the IR stage, the diffusion model refines the subject with a specific style. As shown in Fig. 16 of the revised paper, CoDi generates images with consistent subject appearance and diverse styles. **This effectively addresses the inability of prior methods (e.g., ConsiStory) to handle different styles, as explicitly noted in their limitations.**

---

> > ### Author Response · Authors · 2025-11-27
> >
> > Thank you for your reply and for helping improve our work so far! If you have any further questions, please feel free to raise them during the discussion period.

---

### Official Review · Reviewer_tP1C · 2025-11-01

**Soundness:** 3
**Presentation:** 3
**Contribution:** 3
**Rating:** 6
**Confidence:** 3

**Summary:**

This study introduces a novel training-free framework for subject-consistent image generation. The framework aims to address the limitation of existing methods that sacrifice pose and layout diversity to maintain consistency. Its core design employs a staged processing strategy: during the early generation phase, optimal transport is utilized for coarse-grained identity feature transfer to preserve subject consistency; in the later generation phase, fine-grained refinement is applied to the image features. Experimental results demonstrate that the proposed method outperforms existing mainstream approaches across three key metrics—subject consistency, pose diversity, and text alignment—achieving state-of-the-art performance on public benchmarks.

**Strengths:**

1.The key innovation is the explicit decoupling of identity alignment into coarse-grained transport in early steps and fine-grained refinement in later steps, which is a well-motivated approach based on the progressive nature of diffusion models.
2.A significant strength is the superior balance it achieves. As claimed, the paper provides strong evidence that the method outperforms existing training-free baselines in subject consistency while preserving significantly greater pose diversity and text alignment, addressing a well-known trade-off in the field.

**Weaknesses:**

1.For long-story generation scenarios, it is crucial to maintain consistency in both character identity and their apparel. However, the results presented in the paper demonstrate that the method primarily ensures identity consistency, while the consistency of clothing remains inadequate. In my opinion, this limitation would significantly restrict the method's practicality in long-story applications.

2.The method has a core reliance on binary masks derived from cross-attention maps to extract identity information from the reference image. However, recent and emerging generative models, such as SD3 and other DiT-based architectures, have moved away from using cross-attention mechanisms. This fundamental incompatibility means that the proposed facial extraction technique faces significant challenges in being adapted to these mainstream, state-of-the-art generative models, thereby limiting its generalizability and future relevance.

**Questions:**

1.How is the iterative generation performance? Can it maintain identity consistency?
2.Why does the style of the generated images vary? Since style should also be correlated with features from certain layers of the U-Net, why do the other two models (ConsiStory and StoryDiffusion) not exhibit style changes (based on the observation from Figure 3 in the paper)? Could this limitation affect the model's practical applicability?

---

> ### Author Response · Authors · 2025-11-22
>
> Thank you for your thoughtful comments and suggestions.
>
> **W1:This method prioritizes identity consistency while neglecting clothing consistency, which is equally essential for long-story generation scenarios.**
>
> **A1:**  This issue can be addressed by adding the clothing description in the prompt and adjusting the subject mask threshold to include clothing in the foreground mask . As shown in the Fig.15 of revised paper, CoDi can maintain both subject and clothing consistency while preserving pose diversity for long-story generation.
>
> **W2: Can the method be generalized to new DiT-based models, which do not directly use cross-attention maps and seem unable to extract subject masks?**
>
> **A2:** CoDi can be generalized to DiT-based models. While DiT models lack cross-attention layers and use global self-attention by concatenating text and image tokens, the fundamental attention mechanism remains similar. Subject masks can still be extracted using the attention weights between image and text tokens. We successfully adapted CoDi to DiT-based models (Flux), with generation results in Fig.9 of the revised paper showcasing richer details, subject consistency, and pose diversity.
>
> **Q1: How is iterative generation performance and can it maintain identity consistency?**
>
> **A3:** CoDi effectively maintains identity consistency. It achieves the best performance across all three subject consistency metrics, and extensive experimental examples demonstrate its strong consistency.
>
> **Q2: Why does the style of the generated images vary, while the other two baselines do not, as shown in Figure 3? Could this impact the model's practical applicability?**
>
> **A4:** The observed style variation likely stems from differences in the backgrounds of our results, as CoDi preserves the background of the original target image (generated by Vanilla SDXL). In contrast, the other two baselines maintain subject consistency while easily modifying the background. We argue that subject-consistent generation should prioritize preserving subject identity while minimizing background disruption or influence. Based on our experiments and the additional results presented in the paper, this variation does not affect the model's practical applicability.

---

> > ### Author Response · Authors · 2025-11-27
> >
> > Thank you for your reply and for helping improve our work so far! If you have any further questions, please feel free to raise them during the discussion period.

---

### Author Response · Authors · 2025-12-01
**AC Letter: Summary of Rebuttal & Discussion for Paper#11910**

**Dear Area Chair,**

Due to the recent system revert and the assignment of new ACs, we are writing to provide a concise summary of the consensus reached during the discussion period, particularly regarding the resolution of critical concerns and the resulting score updates.

**1.Score Update (6/6/2 → 6/6/6)**

Our initial scores were 6, 6, and 2. During the discussion, we successfully addressed the concerns of Reviewer VX3M, who explicitly confirmed in their final comment: "I have decided to raise my score to 6".

**2.Resolution with Reviewer VX3M (Score raised 2 → 6)**

Reviewer VX3M initially raised concerns regarding the utility of Optimal Transport (OT) and the **novelty of the Identity Refinement (IR) module**.

- **Resolution on OT:** We **resolved the misunderstanding regarding the OT mechanism**, clarifying that OT acts as a soft attention mechanism that preserves structural alignment better than greedy cosine matching . We also provided empirical evidence showing OT improves pose diversity by **2.29x** compared to cosine similarity.

- **Resolution on Novelty:** We clarified that the **core contribution** lies in the **decoupled strategy** (separating consistency and diversity), rather than the IR module in isolation. We demonstrated that this decoupled approach empirically outperforms coupled methods like ConsiStory. This aligns with Reviewer tP1C, who explicitly praised this decoupling strategy as a **"key innovation"** and a **"well-motivated approach"**.

- **Outcome:** Reviewer VX3M acknowledged the clarification and concluded that **"its novelty remains sufficient for the ICLR bar"** and **"the authors' experiments empirically confirm its effectiveness"**.

**3.Consensus with Reviewers tP1C (Rating 6) and rZBz (Rating 6)**

While these reviewers did not provide further comments during the discussion period, they were initially positive and shared a primary concern regarding the method's generalizability to modern **DiT-based architectures** (e.g., Flux).

- **Resolution:** We successfully adapted CoDi to **Flux (a DiT-based model)** and included these results in the revision (Fig.9). This demonstrated that CoDi is architecture-agnostic.

- **Additional Fixes:**

    - For Reviewer tP1C, we addressed the "clothing consistency" issue in long-story generation by demonstrating that prompt adjustments and mask thresholds effectively maintain apparel consistency.

    - For Reviewer rZBz, we clarified that the "style inconsistency" was due to background preservation from the reference image and showed CoDi effectively decouples subject identity from style.


**4.Summary of Revisions**

To address the feedback from all reviewers, we have made the following key revisions:

- **OT vs. Cosine Similarity:** Provided detailed theoretical analysis and empirical verification to demonstrate the superior utility of Optimal Transport compared to cosine similarity for preserving pose diversity.

- **DiT Adaptation:** Successfully adapted CoDi to the Flux model, demonstrating the method's generalization capabilities beyond U-Net architectures.

- **Consistency & Style Analysis:** Provided additional qualitative results to demonstrate robust clothing consistency in long story generation, and clarified that our method effectively decouples subject identity from style—**overcoming a known limitation of prior methods (e.g., ConsiStory) in handling diverse stylistic scenarios.**

- **Comparison with SOTA training methods:** Added comparative experiments with state-of-the-art training-based models (e.g., UNO) using face-only reference. Results demonstrate that CoDi achieves superior **pose diversity** and **prompt fidelity**, effectively avoiding the "identical layout" issue observed in baselines .

We believe the consensus among all three reviewers is now positive (6/6/6), acknowledging both the novelty and the extensive empirical verification of our method.

Thank you for your time and for managing this challenging situation.

Best regards,

Authors of Paper #11910

---

### Meta-Review · Area_Chair_q6S1 · 2026-01-07

**Summary:**

This paper introduces **CoDi**, a training-free framework for subject-consistent image generation. The method addresses the trade-off between maintaining subject identity and achieving pose diversity by decoupling the process into two stages: "Identity Transport" (utilizing Optimal Transport for coarse alignment) and "Identity Refinement" (for fine-grained feature integration).

During the review process, the reviewers initially raised three main concerns: the method's reliance on U-Net based architectures (SDXL) that raises potential incompatibility with modern DiT-based models (e.g., Flux, SD3); the theoretical justification for using Optimal Transport (OT) over simpler metrics like cosine similarity; and practical limitations in long-story generation, specifically regarding clothing consistency and style variations. Despite these reservations, reviewers acknowledged the paper's strengths in achieving a superior balance between pose diversity and subject consistency without additional training.

**Reviewer Concerns:**

### Addressed by rebuttal
- **Architecture Compatibility**: The authors successfully demonstrated that the method (CoDi) can be adapted to DiT-based models (specifically Flux) by utilizing attention weights between image and text tokens, addressing the concern that the method relied solely on cross-attention maps from U-Net architectures.
- **Theoretical Validity of Optimal Transport (OT)**: The authors provided empirical evidence showing OT significantly outperforms cosine similarity in pose diversity (2.29x improvement). The authors also clarified that the Cost Matrix prevents high-penalty semantic mismatches, effectively functioning as a "soft attention" mechanism, which convinced the dissenting reviewer to raise their score.
- **Clothing Consistency**: Concerns about the method failing to preserve clothing in long-story generation were addressed by new experiments (Fig. 15), showing that adjusting mask thresholds and prompts resolves the issue.
- **Style Variation**: The authors clarified that style variations stemmed from background preservation in the reference images and demonstrated that the method can indeed generate consistent subjects with diverse styles if prompted correctly.

**Reviewer Scores:**

- **Reviewer tP1C**: Score 6 (Marginally above acceptance) -> 6. (The reviewer was already positive, and their specific concerns about DiT compatibility and clothing consistency were fully addressed with the additional visual evidences).
- **Reviewer rZBz**: Score 6 (Marginally above acceptance) -> 6 (This reviewer was positive but concerned about DiT validation; the authors provided the requested Flux adaptation results, satisfying the primary critique).
- **Reviewer VX3M**: Score 2 (Reject) -> Score 6 (Marginally above acceptance). Authors clarified the OT mechanism and the reviewer acknowledged the method's empirical success.

---

### Decision · Program_Chairs · 2026-01-26

Accept (Poster)